# An Exploration-by-Optimization Approach to Best of Both Worlds in Linear Bandits

**Shinji Ito**
NEC Corporation, RIKEN AIP
i-shinji@nec.com

**Kei Takemura**
NEC Corporation
kei_takemura@nec.com

## Abstract

In this paper, we consider how to construct best-of-both-worlds linear bandit algorithms that achieve nearly optimal performance for both stochastic and adversarial environments. For this purpose, we show that a natural approach referred to as exploration by optimization [Lattimore and Szepesvári, 2020b] works well. Specifically, an algorithm constructed using this approach achieves $O(d\sqrt{T \log T})$-regret in adversarial environments and $O(\frac{d^2 \log T}{\Delta_{\min}})$-regret in stochastic environments. Symbols $d$, $T$ and $\Delta_{\min}$ here represent the dimensionality of the action set, the time horizon, and the minimum sub-optimality gap, respectively. We also show that this algorithm has even better theoretical guarantees for important special cases including the multi-armed bandit problem and multitask bandits.

## 1 Introduction

The linear bandit problem [Abernethy et al., 2008] is a fundamental sequential decision-making problem in which a player chooses a $d$-dimensional vector $a_t$ from a given *action set* $\mathcal{A} \subseteq \mathbb{R}^d$ and incurs loss $f_t$. The expectation of $f_t$ is assumed to be expressed as an inner product of $\ell_t$ and $a_t$ with an unknown *loss vector* $\ell_t \in \mathbb{R}^d$ in each round $t = 1, 2, \ldots, T$. This paper focuses on the (expected) regret defined as $R_T = \max_{a^* \in \mathcal{A}} \mathbf{E}[\sum_{t=1}^{T} \langle \ell_t, a - a^* \rangle]$. It is known that the order of achievable regret can vary greatly depending on the structure of the environment that generates losses. For example, in stochastic environments, i.e., if the loss vector $\ell_t$ is fixed at $\ell^*$ for all $t$, there exists an algorithm that achieves regret of $O(\log T)$. By way of contrast, for adversarial environments in which the loss vector changes arbitrarily, the optimal regret bound is $\Theta(\sqrt{T})$ [Abernethy et al., 2008, Bubeck et al., 2012]. While this clearly suggests the importance of choosing an algorithm that is well-suited to the environment, in real-world applications it is often quite difficult, if not virtually impossible, to know the type of environment in advance.

One promising approach to this difficulty is to apply *best-of-both-worlds (BOBW)* algorithms [Bubeck and Slivkins, 2012] that work nearly optimally both for stochastic and adversarial environments. The first BOBW algorithm for linear bandit problems was proposed by Lee et al. [2021]; it achieves $O(\sqrt{dT \log(T) \log(T|\mathcal{A}|)})$-regret in adversarial environments and $O(c^* \log(T|\mathcal{A}|) \log T)$-regret in stochastic environments, where $c^* = c(\mathcal{A}, \ell^*)$ represents an instance-dependent quantity characterizing the instance-optimal asymptotic bound for stochastic environments [Lattimore and Szepesvari, 2017]. This algorithm by Lee et al. [2021] also has the notable advantage that these regret guarantees hold with high probability. On the other hand, their regret bound for stochastic environments includes an extra $\log T$ factor.[1] This issue of extra $\log T$ factors has been resolved in a recent study by Dann et al. [2023b]. They have proposed algorithms achieving regret bounds with optimal dependency on $T$, which are summarized here in Table 1. Their regret bounds for stochastic environments depend

---

[1] As shown in Appendix D by Lee et al. [2021], this extra $\log T$ factor is inevitable as long as we pursue high-probability BOBW regret bounds.

Table 1: Regret bounds for linear bandits

| Reference | Adversarial | Stochastic |
|---|---|---|
| Lattimore and Szepesvari [2017] | | $c^* \log T + o(\log T)$ |
| Bubeck et al. [2012] | $O(\sqrt{dT \log(|\mathcal{A}|)})$ | |
| Lee et al. [2021] | $O\left(\sqrt{dT \log(T) \log(T|\mathcal{A}|)}\right)$ | $O\left(c^* \log(T|\mathcal{A}|) \log T\right)$ |
| Dann et al. [2023b, Cor.7] | $O\left(\sqrt{d^2 T \log T}\right)$ | $O\left(\frac{d^2 \log T}{\Delta_{\min}}\right)$ |
| Dann et al. [2023b, Cor.12] | $O\left(\sqrt{dT \log(|\mathcal{A}|)}\right)$ | $O\left(\frac{d \log(|\mathcal{A}|) \log T}{\Delta_{\min}}\right)$ |
| **[This work, Theorem 1]** | $O\left(\sqrt{\alpha dT \log T}\right)$ | $O\left(\beta d \log T\right)$ |

Table 2: Bounds on parameters $\alpha$ and $\beta$

| | Setting | Action set | $\alpha$ | $\beta$ |
|---|---|---|---|---|
| Corollary 1 | General | $\mathcal{A} \subseteq \mathbb{R}^d$ | $O(d)$ | $O(d/\Delta_{\min})$ |
| Corollary 2 | Multi-armed | $\mathcal{A} = \mathcal{A}_d := \{e_1, \ldots, e_d\}$ | $O(1)$ | $O(1/\Delta_{\min})$ |
| Corollary 3 | Hypercube | $\mathcal{A} = \{0,1\}^d$ | $O(d)$ | $O(1/\Delta_{\min})$ |
| Corollary 3 | Multitask | $\mathcal{A} = \mathcal{A}_{d_1} \times \cdots \times \mathcal{A}_{d_m}$ | $O(m)$ | $O(1/\Delta_{\min})$ |

on the *minimum sub-optimality gap* $\Delta_{\min}$. Note that we have $c^* \leq O(d/\Delta_{\min})$ [Lee et al., 2021, Lemma 16] and that this inequality can be arbitrarily loose [Lattimore and Szepesvari, 2017]. The study by Dann et al. [2023b] involves the proposal of a general reduction technique, which has the remarkable advantage of being applicable to a variety of sequential decision-making problems in addition to linear bandit problems, though it does tend to complicate the structure and implementation of the algorithms.

The main contribution of this paper is to show that BOBW algorithms can be constructed via a (conceptually) simple approach referred to as *exploration by optimization* (EXO) [Lattimore and Szepesvári, 2020b, Lattimore and Gyorgy, 2021, Foster et al., 2022]. In this approach, we update the *reference distribution* $q_t$ using the exponential weight method and then compute sampling distribution $p_t$ for action $a_t$ (by modifying $q_t$ moderately) and loss estimator $g_t$ by solving an optimization problem so that an upper bound on regret is minimized. As shown by Lattimore and Gyorgy [2021] and Foster et al. [2022], this natural approach has a connection to both the information ratio [Russo and Van Roy, 2014] and the decision-estimation coefficient [Foster et al., 2021], and it achieves nearly optimal worst-case regret bounds for some online decision-making problems. These existing studies on EXO, however, focus on adversarial environments, and not much has yet been shown about the potential of its working for stochastic environments. In this paper, we show that the EXO approach also works for stochastic environments to achieve $O(\log T)$-regret.

The regret bounds with the EXO approach and in existing studies are summarized in Table 1. This paper shows the regret bounds of $O(\sqrt{\alpha dT \log T})$ in adversarial environments and of $O(\beta d \log T)$ in stochastic environments, where $\alpha$ is a parameter that depends on $\mathcal{A}$ and $\beta$ is a parameter depending on $\mathcal{A}$ and $\ell^*$. Bounds on $\alpha$ and $\beta$ shown in this paper are summarized in Table 2. For general action sets, the EXO approach reproduces the regret bound shown by Dann et al. [2023b, Corollary 7]. Further, for some special cases, such as multi-armed bandits and hypercube linear bandits, it achieves improved regret bounds that are tight in stochastic regimes parametrized by the sub-optimality gap $\Delta_{\min}$.

To show BOBW regret bounds, we consider here the EXO approach combined with the *continuous exponential weights method* [Hoeven et al., 2018], in which we update reference distributions $q_t$ over a convex set. In our regret analysis of the exponential weight, the ratio of the variance of the sampling distribution $p_t$ to the variance of $q_t$ plays an important role: the larger the former, the better the guarantee obtained. To show $O(\log T)$-regret bounds for stochastic environments, we exploit the fact that $q_t$ is a *log-concave distribution* [Lovász and Vempala, 2007]. Intuitively speaking, as any log-concave distribution has most of its weight concentrated around its mean (see, e.g., [Lovász and Vempala, 2007, Lemma 5.17]), we may choose sampling distribution $p_t$ so that it has a larger variance than $q_t$ which leads to improved regret bounds.

Our regret analysis uses the *self-bounding technique* [Wei and Luo, 2018, Zimmert and Seldin, 2021] to show regret bounds for stochastic environments. One advantage of this technique is that a regret bound also leads to *stochastic environments with adversarial corruptions* or *corrupted stochastic environments* [Lykouris et al., 2018, Li et al., 2019]. As with existing algorithms based on the self-bounding technique, our algorithm achieves a regret bound of $O(\alpha d T \log T + \sqrt{C \alpha d T \log T})$ for corrupted stochastic environments with the corruption level $C \geq 0$. More generally, these regret bounds are achieved for environments in *adversarial regimes with self-bounding constraints* [Zimmert and Seldin, 2021], details of which are provided here in Section 3.

Another contribution of this paper is to establish a connection between the EXO approach and the SCRiBLe algorithm [Abernethy et al., 2008, 2012] (and its extensions [Dann et al., 2023b, Ito and Takemura, 2023]). The SCRiBLe algorithm uses a follow-the-regularized-leader (FTRL) method with self-concordant barrier regularization to select a point in the action set, rather than a probability distribution. Abernethy et al. [2008] have shown that this approach achieves $O(\sqrt{d^3 T})$-regret bounds for adversarial environments. Recently, Dann et al. [2023b] and Ito and Takemura [2023] have shown that it can also achieve $O(\frac{d^3 \log T}{\Delta_{\min}})$-regret in stochastic environments, through modification of the sampling distributions and loss estimators. To better understand the connection between these studies and the EXO approach, we propose a variant that we refer to as mean-oriented EXO and which includes EXO with the continuous exponential weight method as a special case. Given this correspondence, existing SCRiBLe-type algorithms can be interpreted as EXO-based methods in which the optimization is roughly solved with approximation ratios $\geq \Omega(d)$.

The results of this paper establish a connection between the framework of exploration by optimization and best-of-both-worlds regret guarantees. We would like to emphasize that there is potential to extend this result to a broader class of sequential decision-making beyond linear bandits, such as partial monitoring problems and episodic MDPs. In fact, the EXO framework can be applied to a general model of sequential decision-making with structured observation [Foster et al., 2022] and is known to have a connection to several key concepts in sequential decision-making, such as the Decision-Estimation Coefficients [Foster et al., 2021] and information-directed sampling [Lattimore and Gyorgy, 2021, Russo and Van Roy, 2014], which has been extensively applied to partial monitoring [Lattimore and Szepesvári, 2020b] and linear partial monitoring [Kirschner et al., 2020]. Building further relationships between these generic frameworks and BOBW regret analysis will be an important topic to work on in the future.

## 2 Related work

For adversarial linear bandit problems, various approaches have been considered, including algorithms based on self-concordant barriers (SCRiBLe) [Abernethy et al., 2008, 2012, Rakhlin and Sridharan, 2013], discrete exponential weights (Geometric Hedge) [Dani et al., 2008, Bubeck et al., 2012, Cesa-Bianchi and Lugosi, 2012], and continuous exponential weights (continuous Geometric Hedge) [Hazan and Karnin, 2016, Bubeck and Eldan, 2019], all of which can be interpreted as methods in the FTRL framework. We can see that continuous exponential weight methods can be interpreted as an FTRL approach with an entropic barrier, a $d$-self-concordant barrier [Chewi, 2021], which implies a close relationship between the SCRiBLe algorithm and the continuous Geometric Hedge, as Bubeck and Eldan [2019] have noted. One key difference lies in how the sampling distribution $p_t$ is chosen; the SCRiBLe algorithm employs a sampling scheme supported on the Dikin ellipsoid while the continuous Geometric Hedge designs $p_t$ by mixing the reference distribution $q_t$ with some exploration basis (e.g., a G-optimal design distribution). As a result of this difference, the latter can achieve an $O(d^{-1/2})$-times better regret bound than the former. Recent studies by Dann et al. [2023b, Theorem 3] and Ito and Takemura [2023] have proposed new sampling schemes based on Dikin ellipsoid sampling to achieve BOBW regret bounds. This paper takes these approaches a step further, determining a sampling distribution itself on the basis of an optimization problem for minimizing regret, which leads to improved performance.

The self-bounding technique [Wei and Luo, 2018, Zimmert and Seldin, 2021] is known as a promising approach for designing and analyzing BOBW algorithms. The greatest impetus for the widespread use of this technique is considered to come from the Tsallis-INF algorithm [Zimmert et al., 2019] for the multi-armed bandit problem. Since the study on Tsallis-INF, FTRL methods with Tsallis-entropy regularizer (with some modification) have been used successfully in a variety of sequential

decision-making problems with limited (structured) observation, including combinatorial semi-bandits [Zimmert et al., 2019], bandits with switching costs [Rouyer et al., 2021, Amir et al., 2022], bandits with delayed feedback [Masoudian et al., 2022], online learning with feedback graphs [Erez and Koren, 2021, Rouyer et al., 2022]. dueling bandits [Saha and Gaillard, 2022], and episodic MDPs [Jin and Luo, 2020, Jin et al., 2021, Dann et al., 2023a]. On the other hand, a few (rather limited number of) studies have constructed BOBW algorithms using regularizer functions other than the Tsallis entropy. For example, Ito [2021] proposed a BOBW multi-armed bandit algorithm by using log-barrier regularization equipped with adaptive learning rates. More recently, Shannon-entropy regularization with some specific adjustment methods for learning rates has been shown to provide BOBW regret guarantees for several problems including online learning with feedback graphs [Ito et al., 2022b], partial monitoring [Tsuchiya et al., 2023], and linear bandits [Kong et al., 2023]. It seems that these Shannon-entropy-based methods have the strength of being versatile while these have the weakness that their regret bounds for stochastic environments include extra $O(\log T)$-factors. The approach in this paper may appear to be related to the previous studies using the Shannon entropy [Ito et al., 2022b, Tsuchiya et al., 2023, Kong et al., 2023] as it employs exponential weight methods, but it is more related to the studies using log-barrier regularization [Ito, 2021]. In fact, both log barriers and continuous exponential weight can be captured in the category of self-concordant barriers as we will discuss in Sections 5 and 6. These also have in common the type of regret bounds: $O(\log T)$ in stochastic environments and $O(\sqrt{T \log T})$ in adversarial environments.

## 3 Problem setting

In linear bandit problems, the player is given an *action set* $\mathcal{A} \subseteq \mathbb{R}^d$, which is a closed bounded set of $d$-dimensional vectors. Without loss of generality, we assume that $\mathcal{A}$ spans all vectors in $\mathbb{R}^d$. In each round $t$, the environment chooses a *loss vector* $\ell_t \subseteq \mathbb{R}^d$ while the player selects an action $a_t \in \mathcal{A}$ without knowing $\ell_t$. The player then gets feedback of $f_t \in [-1, 1]$ that is drawn from a distribution $M_{a_t}$, where the series of distributions $M = (M_a)_{a \in \mathcal{A}}$ is in $\mathcal{M}_{\ell_t} := \{(M_a)_{a \in \mathcal{A}} | \forall a \in \mathcal{A}, \ \mathbf{E}_{f \sim M_a}[f] = \langle \ell_t, a \rangle\}$. The condition that the support of $M_a$ is included in $[-1, 1]$ for all $a \in \mathcal{A}$ implies that $\langle \ell_t, a \rangle \in [-1, 1]$ holds for all $a \in \mathcal{A}$ and $t \in [T]$. In other words, it is assumed that $\ell_t \in \mathcal{L}$ for any $t$, where we define $\mathcal{L} \subseteq \mathbb{R}^d$ by $\mathcal{L} = \{\ell \in \mathbb{R}^d \mid \forall a \in \mathcal{A}, \ |\langle \ell, a \rangle| \leq 1\}$. The performance of the player is evaluated by means of *regret* defined as

$$R_T(a^*) = \mathbf{E}\left[\sum_{t=1}^{T} \langle \ell_t, a_t \rangle - \sum_{t=1}^{T} \langle \ell_t, a^* \rangle\right], \quad R_T = \sup_{a^* \in \mathcal{A}} R_T(a^*). \tag{1}$$

**Regimes of environments** In *stochastic environments*, $\ell_t$ is assumed to be round-invariant, i.e., there exists $\ell^* \in \mathcal{L}$ such that $\ell_t = \ell^*$ holds for all $t$. In *adversarial environments*, $\ell_t$ can be chosen in an adversarial manner depending on the history so far $((\ell_s, a_s))_{s=1}^{t-1}$. *Stochastic environments with adversarial corruptions* correspond to an intermediate setting between these two types of environments. These environments are parametrized by the *corruption level* $C \geq 0$, in which the loss vectors $(\ell_t)$ are assumed to satisfy $\mathbf{E}\left[\sum_{t=1}^{T} \max_{a \in \mathcal{A}} |\langle \ell_t - \ell^*, a \rangle|\right] \leq C$. To capture these settings of environments, we define an *adversarial regime with a $(\ell^*, a^*, C, T)$-self-bounding constraint* [Zimmert and Seldin, 2021] in which the environments choose losses so that

$$R_T(a^*) \geq \mathbf{E}\left[\sum_{t=1}^{T} \langle \ell^*, a_t - a^* \rangle\right] - C \tag{2}$$

holds for any algorithms. This regime includes (corrupted) stochastic environments. In fact, any corrupted stochastic environments with the true loss $\ell^* \in \mathcal{L}$ and the corruption level $C \geq 0$ are included in an adversarial regime with a $(\ell^*, a^*, C, T)$-self-bounding constraint. Our regret analysis for (corrupted) stochastic environments relies only on the condition of (2), as in some existing studies [Zimmert and Seldin, 2021, Dann et al., 2023b].

**Reduction to convex action set** Let $\mathcal{X} = \text{conv}(\mathcal{A})$ represent the convex hull of $\mathcal{A}$. We may consider the action set $\mathcal{X}$ instead of $\mathcal{A}$, i.e., if we can construct an algorithm $\text{Alg}_{\mathcal{X}}$ for the problem with action set $\mathcal{X}$, we can convert it to an algorithm $\text{Alg}_{\mathcal{A}}$ for the problem with $\mathcal{A}$. In fact, for any

output $a_t' \in \mathcal{X}$ of $\mathtt{Alg}_{\mathcal{X}}$, we can pick $a_t \in \mathcal{A}$ so that $\mathbf{E}[a_t|a_t'] = a_t'$.[2] When defining $a_t$ as the output of $\mathtt{Alg}_{\mathcal{A}}$, we have $\mathbf{E}[f_t|a_t'] = \mathbf{E}[\langle \ell_t, a_t \rangle|a_t'] = \langle \ell_t, a_t' \rangle$, which means that the expectation of the feedback $f_t$ given $a_t'$ is indeed a linear function in $a_t'$ and that the regret for $\mathtt{Alg}_{\mathcal{A}}$ coincides with that for $\mathtt{Alg}_{\mathcal{X}}$.

# 4 Exploration by optimization

## 4.1 Algorithm framework and regret analysis

Let $\mathcal{P}(\mathcal{X})$ denote the set of all distributions over $\mathcal{X}$. For any distribution $p \in \mathcal{P}(\mathbb{R}^d)$, denote $\mu(p) = \mathbf{E}_{x \sim p}[x] \in \mathbb{R}^d$, $H(p) = \mathbf{E}_{x \sim p}[xx^\top] \in \mathbb{R}^{d \times d}$, and $V(p) = H(p) - \mu(p)\mu(p)^\top \in \mathbb{R}^{d \times d}$. We consider algorithms that choose $a_t$ following a *sampling distribution* $p_t \in \mathcal{P}(\mathcal{X})$ in each round $t$. To determine $p_t$, we compute an (approximately) unbiased estimator $g_t(x; a_t, f_t)$ of the loss function $\langle \ell_t, x \rangle$. Let $\mathcal{G}$ be the set of all estimators $g$ that are linear in the first variable, i.e., $\mathcal{G} = \{g : \mathcal{X} \times \mathcal{X} \times [-1,1] \to \mathbb{R} \mid \exists h : \mathcal{X} \times [-1,1] \to \mathbb{R}^d, \ g(x; a, f) = \langle h(a,f), x \rangle\}$. For any distribution $p \in \mathcal{P}(\mathcal{X})$, let $\mathcal{G}^{\mathrm{u}}(p) \subseteq \mathcal{G}$ represent the set of all unbiased estimators:

$$\mathcal{G}^{\mathrm{u}}(p) = \left\{ g \in \mathcal{G} \mid \forall \ell \in \mathcal{L}, \forall x \in \mathcal{X}, \forall M \in \mathcal{M}_\ell, \quad \mathop{\mathbf{E}}_{a \sim p, f \sim M_a}[g(x; a, f)] = \langle \ell, x \rangle \right\}. \quad (3)$$

A typical choice of $g$ given $p$ is the estimator defined as follows:

$$g(x; a, f) = \langle h(a, f), x \rangle, \quad \text{where} \quad h(a, f) = f \cdot (H(p))^{-1} a, \quad (4)$$

where $H(p)$ is assumed to be non-singular. This is an unbiased estimator, i.e., $g$ defined by (4) is an element of $\mathcal{G}^{\mathrm{u}}$. In fact, if $M \in \mathcal{M}_\ell$, we have $\mathbf{E}_{a \sim p, f \sim M_a}[h(a, f)] = \mathbf{E}_{a \sim p}[(H(p))^{-1} aa^\top \ell] = (H(p))^{-1} H(p) \ell = \ell$.

Using $\{g_s\}_{s=1}^t$, we set a *reference distribution* $q_t$ over $\mathcal{X}$ by the continuous exponential weights method as follows: $q_t(x) \propto q_0(x) \exp(-\eta_t \sum_{s=1}^{t-1} g_s(x; a_s, f_s))$, where $q_0$ is an arbitrary initial distribution over $\mathcal{X}$ and $\eta_t$ is a learning rate such that $\eta_1 \geq \eta_2 \geq \cdots \geq \eta_T > 0$. We may choose $\eta_t$ depending on the historical actions and observations so far $((a_s, f_s))_{s=1}^{t-1}$. Let $D_{\mathrm{KL}}(q', q)$ represent the Kulllback-Leibler divergence of $q'$ from $q$. Note that the exponential weights method can be interpreted as a follow-the-regularized-leader approach over $\mathcal{P}$ with the regularizer function $q \mapsto D_{\mathrm{KL}}(q, a_0)$, i.e., the distribution $q_t$ is a solution to the following optimization problem:

$$q_t \in \mathop{\arg\min}_{q \in \mathcal{P}(\mathcal{X})} \left\{ \mathop{\mathbf{E}}_{x \sim q} \left[ \sum_{s=1}^{t-1} g_s(x; a_s, f_s) \right] + \frac{1}{\eta_t} D_{\mathrm{KL}}(q, q_0) \right\}. \quad (5)$$

We may determine the sampling distribution $p_t$ on the basis of $q_t$ in an arbitrary way. The regret is then bounded as follows:

**Lemma 1.** *For any distribution $p^* \in \mathcal{P}(\mathcal{X})$ such that $D_{\mathrm{KL}}(p^*, q_0) < \infty$, the expected value of the regret $R_T(a^*)$ for $a^* \sim p^*$ is bounded by*

$$\mathbf{E} \left[ \sum_{t=1}^T \left( \langle \ell_t, \mu(p_t) - \mu(q_t) \rangle + \mathrm{bias}(g_t, p_t, q_t, \ell_t) + \frac{1}{\eta_t} \mathrm{stab}(\eta_t g_t(\cdot; a_t, f_t), q_t) \right) + \frac{D_{\mathrm{KL}}(p^*, q_0)}{\eta_{T+1}} \right],$$

*where* $\quad \mathrm{bias}(g, p, q, \ell) = \sup_{a^* \in \mathcal{X}, M \in \mathcal{M}_\ell} \left\{ \mathop{\mathbf{E}}_{a \sim p, x \sim q, f \sim M_a}[\langle \ell, x - a^* \rangle - g(a^*; a, f) + g(x; a, f)] \right\},$

$$\mathrm{stab}(g, q) = \sup_{q' \in \mathcal{P}(\mathcal{X})} \left\{ \mathop{\mathbf{E}}_{x \sim q}[g(x)] - \mathop{\mathbf{E}}_{x \sim q'}[g(x)] - D_{\mathrm{KL}}(q', q) \right\}. \quad (6)$$

This follows from the standard analysis for exponential update methods or FTRL (see, e.g., [Lattimore and Szepesvári, 2020a, Exercise 28.12]), and all omitted proofs can be found in the appendix. For the case of the exponential weights, the *stability term* $\mathrm{stab}(g, q)$ can be bounded as

$$\mathrm{stab}(g, q) \leq \inf_{\bar{g} \in \mathbb{R}} \mathop{\mathbf{E}}_{x \sim q}[\phi(g(x) - \bar{g})], \quad \text{where} \quad \phi(y) = \exp(-y) + y - 1. \quad (7)$$

---

[2]Such a distribution of $a_t$ given $a_t'$ can be computed efficiently e.g., by the algorithm given in [Schrijver, 1998, Corollary 14.1g] or [Mirrokni et al., 2017].

The *bias term* $\mathrm{bias}(g, p, q, \ell)$ corresponds to the bias of the estimater $g(\cdot; a, f)$ for the fucntion $\langle \ell, \cdot \rangle$, under the condition that $a \sim p$, and $\mathbf{E}[f|a] = \langle a, \ell \rangle$. We can easily see that $g \in \mathcal{G}^{\mathrm{u}}(p)$ implies $\mathrm{bias}(g, p, q, \ell) = 0$ for all $q \in \mathcal{P}(\mathcal{X})$ and $\ell \in \mathcal{L}$.

For any reference distribution $q \in \mathcal{P}(\mathcal{X})$ and learning rate $\eta > 0$, we define $\Lambda_{q,\eta}(p, g)$ by

$$\Lambda_{q,\eta}(p, g) = \sup_{\ell \in \mathcal{L}, M \in \mathcal{M}_\ell} \left\{ \langle \ell, \mu(p) - \mu(q) \rangle + \mathrm{bias}(g, p, q, \ell) + \frac{1}{\eta} \mathop{\mathbf{E}}_{a \sim p, f \sim M_a} [\mathrm{stab}(\eta g(\cdot; a, f), q)] \right\}.$$

From Lemma 1, the regret is bounded as

$$\mathop{\mathbf{E}}_{a^* \sim p^*} [R_T(a^*)] \le \mathbf{E} \left[ \sum_{t=1}^{T} \Lambda_{q_t, \eta_t}(p_t, g_t) + \frac{1}{\eta_{T+1}} D_{\mathrm{KL}}(p^*, q_0) \right]. \tag{8}$$

In the *exploration-by-optimization (EXO)* approach [Lattimore and Szepesvári, 2020b, Lattimore and Gyorgy, 2021], we choose $g_t$ and $p_t$ so that the value of $\Lambda_{q_t, \eta_t}(g_t, p_t)$ is as small as possible, i.e., we consider the following optimization problem:

$$\text{Minimize} \quad \Lambda_{q_t, \eta_t}(p, g) \quad \text{subject to} \quad p \in \mathcal{P}(\mathcal{X}), \ g \in \mathcal{G}. \tag{9}$$

We denote the optimal value of this problem by $\Lambda^*_{q_t, \eta_t}$.

**Remark 1.** In most linear bandit algorithms based on exponential weights [Dani et al., 2008, Bubeck et al., 2012, Cesa-Bianchi and Lugosi, 2012, Hazan and Karnin, 2016, Besbes et al., 2019], $p_t$ is defined as $q_t$ mixed with a small component of *exploration basis* $\pi_0$ and $g_t$ is given by (4) with $p = p_t$. The exploration basis $\pi_0$ here is a distribution over $\mathcal{X}$ such that $G(\pi_0) := \sup_{a \in \mathcal{X}} a^\top H(\pi_0) a$ is bounded. It is known that any action set $\mathcal{X} \subseteq \mathbb{R}^d$ admits an exploration basis $\pi_0$ such that $G(\pi_0) \le d$, which is called the G-optimal design or the John's ellipsoid exploration basis. Given such a $\pi_0$, we set $p_t = (1 - \gamma_t) q_t + \gamma_t \pi_0$ with some $\gamma_t \in (0, 1)$. Under the condition of $\eta_t = \Omega(1/d)$ and $\gamma_t = \Theta(d\eta_t)$, we have $\Lambda_{q_t, \eta_t}(p_t, g_t) = O(\gamma_t + \eta_t d) = O(\eta_t d)$, which implies that $\Lambda^*_{q, \eta_t} = O(\eta_t d)$ holds for any $q \in \mathcal{P}(\mathcal{X})$. Hence, from (8), if $D_{\mathrm{KL}}(p^*, q_0) \le B$, by setting $\eta_t = \eta = \Theta(\sqrt{B/(dT)})$, we obtain $\mathbf{E}_{a^* \sim p^*}[R_T(a^*)] = O(\sqrt{BdT})$.

## 4.2 Sufficient condition for best-of-both-worlds regret guarantee

In this section, we show that a certain type of upper bounds on $\Lambda^*_{q, \eta}$ that depend on reference distribution $q$ will lead to best-of-both-worlds regret bounds. In the following, we set the initial disturibution $q_0$ to be a uniform distribution over $\mathcal{X}$. Denote $\Delta_{\ell^*}(x) = \langle \ell^*, x - a^* \rangle$ for any $\ell^* \in \mathcal{L}$. The following lemma provides a sufficient condition for best-of-both-worlds regret guarantees:

**Theorem 1.** *Suppose that there exist $\alpha > 0, \beta > 0$ and $\eta_0 > 0$ such that*

$$\Lambda^*_{q_t, \eta} \le \eta \cdot \min\{\alpha, \beta \cdot \Delta_{\ell^*}(\mu(q_t))\} \tag{10}$$

*for all $t$ and $\eta \le \eta_0$. Then, an algorithm based on the EXO approach achieves the following regret bounds simultaneously: (i) In adversarial environments, we have $R_T = O\left(\sqrt{\alpha dT \log T} + \frac{d \log T}{\eta_0}\right)$.*

*(ii) In environments satisfying (2), we have $R_T = O\left(\left(\beta + \frac{1}{\eta_0}\right) d \log T + \sqrt{C\beta d \log T}\right)$.*

In the proof of this theorem, we consider the algorithm that chooses $(p_t, g_t)$ to be an optimal solution to (9) and that sets learning rates $\eta_t$ by $\eta_t = \min\left\{\eta_0, \sqrt{\frac{d \log T}{\alpha + \sum_{s=1}^{t-1} \eta_s^{-1} \Lambda^*_{q_s, \eta_s}}}\right\}$. A complete proof is given in the appendix.

**Remark 2.** In the implementation of the algorithm, we do not necessarily need to solve the optimization problem (9) exactly. In fact, to achieve the regret upper bounds in Theorem 1, it is sufficient to compute $p_t, g_t$, and $z_t > 0$ that satisfy $\eta_t^{-1} \Lambda_{q_t, \eta_t}(p_t, g_t) \le z_t \le \min\{\alpha, \beta \cdot \Delta_{\ell^*}(\mu(q_t))\}$ for some $\alpha > 0$ and $\beta > 0$, under the assumption of $\eta_t \le \eta_0$. In this case, setting $\eta_t = \min\{\eta_0, \sqrt{d \log T}/\sqrt{\alpha + \sum_{s=1}^{t-1} z_s}\}$ will work.

## 4.3 Regret bounds for concrete examples

In this section, we will see how we can bound $\Lambda^*_{q_t,\eta_t}$. For $q \in \mathcal{P}(\mathcal{X})$, define $\omega(q), \omega'(q) \geq 0$ by

$$\omega(q) = \min_{p \in \mathcal{P}(\mathcal{X}): \mu(p)=\mu(q)} \{H(p)^{-1} \bullet V(q)\}, \tag{11}$$

$$\omega'(q) = \min_{p \in \mathcal{P}(\mathcal{X}): \mu(p)=\mu(q)} \min\{y > 0 \mid d \cdot V(q) \preceq y \cdot H(p)\}. \tag{12}$$

Note that $\omega(q) \leq \omega'(q)$ holds. In fact, from the definition of $\omega'(q)$, there exists $p \in \mathcal{P}(\mathcal{X})$ satisfying $\mu(p) = \mu(q)$ and $d \cdot V(q) \preceq \omega'(q) \cdot H(p)$. For such a distribution $p$, we have $H(p)^{-1} \bullet V(q) = \mathrm{tr}(H(p)^{-1/2}V(q)H(p)^{-1/2}) \leq \mathrm{tr}(d^{-1} \cdot \omega'(q) \cdot I) = \omega'(q)$, which implies that $\omega(q) \leq \omega'(q)$.

Using these, we can construct a bound $\Lambda^*_{q,\eta}$ as the following lemma provides:

**Lemma 2.** *Suppose that $q$ is a log-concave distribution. If $\eta \leq 1/d$, we then have $\Lambda^*_{q,\eta} = O(\eta\omega(q))$. If $\eta \leq 1$, we then have $\Lambda^*_{q,\eta} = O(\eta\omega'(q))$.*

Bounds on $\Lambda^*_{q,\eta}$ in this lemma can be achieved by $p = \gamma\tilde{p} + (1-\gamma)\pi_0$ and $g$ defined by (4), where $\tilde{p}$ is the minimizer on the right-hand side of (11) or (12), $\pi_0$ is a G-optimal design for $\mathcal{X}$, and some mixing weight parameter $\gamma \in (0,1)$. The proof of this lemma is given in the appendix.

**General action set** We can provide an upper bound on $\omega(q)$ by exploiting the property of *log-concave distributions*. A distribution $q \in \mathcal{P}(\mathcal{X})$ is called a log-concave distribution if it has a distribution function that is proportional to $\exp(-h(x))$ for some convex function $h$. Note that the reference distribution $q_t$ is a log-concave function.

**Lemma 3.** *Suppose that $q$ is a log-concave distribution. Let $\ell^* \in \mathcal{L}$ be an arbitrary loss vector such that $a^* \in \arg\min_{a \in \mathcal{A}} \langle \ell^*, a \rangle$ is unique. Denote $\Delta_{\ell^*,\min} = \min_{a \in \mathcal{A} \setminus \{a^*\}} \Delta_{\ell^*}(a)$. We then have $\omega'(q) = O(d \cdot \min\{1, \Delta_{\ell^*}(\mu(q))/\Delta_{\ell^*,\min}\})$.*

We here provide a proof sketch for this lemma. We consider a distribution $p \in \mathcal{P}(\mathcal{A})$ of $a \in \mathcal{A}$ generated by the following procedure: (i) Pick $x \in \mathcal{X}$ following $q$. (ii) Let $\xi$ be an arbitrary distribution over $\mathcal{A}$ such that $\mu(\xi) = x$. In other words, compute $\xi : \mathcal{A} \to \mathbb{R}_{\geq 0}$ such that $\sum_{a' \in \mathcal{A}} \xi(a') = 1$ and $\sum_{a' \in \mathcal{A}} \xi(a')a' = x$. (iii) Pick $a \in \mathcal{A}$ following $\xi$. It is then clear that $\mu(p) = \mu(q)$ holds. Further, it follows from the assumption that $q$ is a log-concave distribution that $(1 - p(a^*))u^\top V(p)u = \Omega(u^\top V(q)u)$ holds for any $a^* \in \mathcal{A}$ and any $u \in \mathbb{R}^d \setminus \{0\}$, which implies $d \cdot V(q) \preceq O((1 - p(a^*))d) \cdot V(p) \preceq O((1-p(a^*))d)H(p)$. Finally, since we have $(1-p(a^*)) \leq \Delta_{\ell^*}(\mu(p))/\Delta_{\ell^*,\min}$, the bound on $\omega'(q)$ in Lemma 3 follows. A full proof can be found in the appendix.

From Lemmas 2 and 3, we obtain the following bounds on $\alpha$ and $\beta$:

**Corollary 1.** *For arbitrary action set $\mathcal{A}$, the condition (10) in Theorem 1 holds with $\alpha = O(d)$, $\beta = O(d/\Delta_{\ell^*,\min})$, and $\eta_0 = O(1)$.*

Hence, there exists an algorithm that achieves $R_T = O(d\sqrt{T \log T})$ in adversarial environments and $R_T = O(\frac{d^2}{\Delta_{\ell^*,\min}} \log T + \sqrt{\frac{Cd^2}{\Delta_{\ell^*,\min}} \log T})$ in stochastically constrained adversarial environments.

**Multi-armed bandits** Suppose a problem setting in which the action set is the canonical bases: $\mathcal{A} = \mathcal{A}_d := \{e_1, e_2, \ldots, e_d\} = \{a \in \{0,1\}^d \mid \|a\|_1 = 1\}$. For this action set, we have the following bound on $\omega(q)$:

**Lemma 4.** *If $\mathcal{A} = \mathcal{A}_d$, for any log-concave distribution $q$, $\omega(q)$ defined in (11) is bounded as $\omega(q) \leq O\left(\sum_{i=1}^d \mu_i(q)(1 - \mu_i(q))\right) \leq O(\min_{1 \leq i \leq d}\{1 - \mu_i(q)\})$.*

Let $\ell^* \in \mathcal{L}$ be an a loss vector such that $e_{i^*} \in \arg\min_{1 \leq i \leq d} \ell_i^*$ is unique. We then have $\Delta_{\ell^*}(\mu) = \sum_{i \neq i^*} \Delta_{\ell^*}(e_i)\mu_i \geq \Delta_{\ell^*,\min}(1 - \mu_{i^*})$, which, combined with Lemma 4, implies $\omega(q) \leq \Delta_{\ell^*}(\mu(q))/\Delta_{\ell^*,\min}$. From this and Lemma 2, we have the following bounds on $\alpha$ and $\beta$:

**Corollary 2.** *For the multi-armed bandit problem $\mathcal{A} = \mathcal{A}_d$, the condition (10) in Theorem 1 holds with $\alpha = O(1)$, $\beta = O(1/\Delta_{\ell^*,\min})$ and $\eta_0 = O(1/d)$.*

**Product-space action sets**  Suppose that $\mathcal{A} \subseteq \mathbb{R}^d$ is expressed as a product space of $m$ sets, i.e., $\mathcal{A} = \mathcal{A}^{(1)} \times \mathcal{A}^{(2)} \times \cdots \times \mathcal{A}^{(m)}$, where $\mathcal{A}^{(j)} \subseteq \mathbb{R}^{d_j}$ for each $j$ and $\sum_{j=1}^{m} d_j = d$. Denote $\mathcal{X}^{(j)} = \mathrm{conv}(\mathcal{A}^{(j)})$. This setting is referred to as the *multi-task bandit* problem [Cesa-Bianchi and Lugosi, 2012], in which the player chooses $a_t^{(j)} \in \mathcal{A}^{(j)}$ for each $j$ in parallel, and then gets only a single feedback corresponding to the sum of the losses for all $m$ actions. For such an action set, probability distribution $q_t$ given by continuous exponential weights can be expressed as a product measure of $m$ log-concave measures $q_t^{(j)} \in \mathcal{P}(\mathcal{X}^{(j)})$. In fact, if a probability density function is expressed as $q(x) = c \exp(\langle \eta L, x \rangle)$ for some $c > 0$ $\eta > 0$, and $L = (L^{(1)}, \ldots, L^{(m)}) \in \mathbb{R}^d$, it can be decomposed as $q(x) = c \exp(\sum_{j=1}^{m} \langle \eta L^{(j)}, x^{(j)} \rangle) = \prod_j^m c^{(j)} \exp(\langle \eta L^{(j)}, x^{(j)} \rangle) = \prod_j^m q^{(j)}(x^{(j)})$, where $x^{(j)} \in \mathcal{X}^{(j)}$ and $c^{(j)} = \int_{x \in \mathcal{X}^{(j)}} \exp(\langle \eta L^{(j)}, x^{(j)} \rangle) \mathrm{d}x$. This fact leads to the following bounds on $\omega(q)$ and $\Lambda_{q,\eta}^*$:

**Lemma 5.** *If $\mathcal{A} = \mathcal{A}^{(1)} \times \cdots \times \mathcal{A}^{(m)}$ and $q \in \mathcal{P}(\mathcal{X})$ can be expressed as a product measure of $q^{(j)} \in \mathcal{P}(\mathcal{X}^{(j)})$ for $j = 1, \ldots, m$, we have $\omega(q) \leq \sum_{j=1}^{m} \omega(q^{(j)})$.*

**Corollary 3.** *If $\mathcal{A} = \mathcal{A}_{d_1} \times \cdots \times \mathcal{A}_{d_m}$, the condition* (10) *in Theorem 1 holds with $\alpha = O(m)$, $\beta = O(1/\Delta_{\ell^*,\min})$, and $\eta_0 = O(1/d)$. If $\mathcal{A} = \{0,1\}^d$, the condition* (10) *in Theorem 1 holds with $\alpha = O(d)$, $\beta = O(1/\Delta_{\ell^*,\min})$, and $\eta_0 = O(1/d)$.*

### 4.4  Computational complexity

At this time, it is not known if there is an efficient algorithm to solve the optimization problems (9). However, it is worth noting that the optimization problem corresponding to the right-hand side of (11) is a convex optimization problem. Further, as the minimum can be achieved by $p \in \mathcal{P}(\mathcal{A})$, we can reduce the problem into a convex optimization problem over $|\mathcal{A}|$-dimensional space, which can be solved efficiently if $|\mathcal{A}|$ is not very large.

Note that it is not always necessary to solve the optimization problem (9) exactly, as mentioned in Remark 2. For example, we can implement an algorithm achieving regret bounds of Corollary 1 given a separation oracle (or equivalently, a linear optimization oracle) for $c\mathcal{A}$, without solving (9) exactly. Indeed, to achieve the regret bounds of Corollary 1, it suffices to find $p$ and $g$ such that $\Lambda_{q_t,\eta_t}(p,g)$ is bounded by the RHS of (10). The construction of such $p$ and $g$ is provided in the proof of Lemmas 2 and 3, which can be performed using a separation oracle for $\mathcal{A}$. In fact, we can obtain samples from $p$ by using the techniques for log-concave sampling (e.g., [Lovász and Vempala, 2007]) and for computing convex combination expression (cf. Carathéodory's theorem for convex hull and [Schrijver, 1998, Corollary 14.1g]). However, the analysis of log-concave sampling and calculations of $H(p)$ (which is required for constructing $g$) including consideration of calculation errors can be highly complicated, and the computational cost can be very large, although on the order of polynomials (e.g., [Hazan and Karnin, 2016, Corollary 6.2]).

## 5  Mean-oriented FTRL and SCRiBLe

For linear bandit problems, we have an alternative approach that we referred to as *mean-oriented FTRL*, in which we compute a *reference mean* $w_t$ given as follows:

$$w_t \in \arg\min_{w \in \mathcal{X}} \left\{ \left\langle \sum_{s=1}^{t-1} h_s(a_s, f_s), w \right\rangle + \frac{1}{\eta_t} \psi(w) \right\}, \tag{13}$$

where $h_s(a_s, f_s)$ is an estimator of $\ell_s$ and $\psi$ is a convex regularization function over $\mathcal{X}$ such that $\min_{x \in \mathcal{X}} \psi(x) = 0$. Let $D_\psi(x', x)$ denote the Bregman divergence associated with $\psi$: $D_\psi(x', x) = \psi(x') - \psi(x) - \langle \nabla \psi(x), x' - x \rangle$. This approach ensures the following regret bound:

**Lemma 6.** *For any point $x^* \in \mathcal{X}$, we have*

$$R_T(x^*) \leq \mathbf{E}\left[\sum_{t=1}^{T}\left(\langle \ell_t, \mu(p_t) - w_t\rangle + \mathrm{bias}(h_t, p_t, w_t, \ell_t) + \frac{\mathrm{stab}_\psi(\eta_t h_t(a_t, f_t), w_t)}{\eta_t}\right) + \frac{\psi(x^*)}{\eta_{T+1}}\right],$$

*where* $\mathrm{bias}(h, p, w, \ell) = \sup_{a^* \in \mathcal{X}, M \in \mathcal{M}_\ell}\left\{\mathop{\mathbf{E}}_{a \sim p, f \sim M_a}[\langle \ell - h(a, f), w - a^*\rangle]\right\},$

$$\mathrm{stab}_\psi(\hat{\ell}, w) = \sup_{w' \in \mathcal{X}}\left\{\left\langle \hat{\ell}, w - w'\right\rangle - D_\psi(w', w)\right\}. \tag{14}$$

One example of linear bandit algorithms based on mean-oriented FTRL is the SCRiBLe algorithm [Abernethy et al., 2008, 2012], which employs *self-concordant barriers* as regularizer functions.

We can also consider an EXO approach for mean-oriented FTRL. For any reference point $w \in \mathcal{X}$ and learning rate $\eta > 0$, define

$$\Lambda_{w,\eta}(p, h) = \sup_{\ell \in \mathcal{L}, M \in \mathcal{M}_\ell}\left\{\langle \ell, \mu(p) - w\rangle + \mathrm{bias}(h, p, w, \ell) + \frac{1}{\eta}\mathop{\mathbf{E}}_{a \sim p, f \sim M_a}[\mathrm{stab}_\psi(\eta h(a, f), q)]\right\}$$

and $\Lambda_{w,\eta}^* = \inf_{p \in \mathcal{P}(\mathcal{X}), \hat{\ell}: \mathcal{X} \times \mathbb{R} \to \mathbb{R}^d} \Lambda_{w,\eta}(p, \hat{\ell})$.

This achieves the following BOBW regret bounds, under assumptions on $\Lambda_{w,\eta}^*$:

**Lemma 7.** *Suppose that $\psi$ is a $\nu$-self-concordant barrier. Suppose that there exist $\alpha > 0, \beta > 0$ and $\eta_0 > 0$ such that*

$$\Lambda_{w_t,\eta}^* \leq \eta \cdot \min\{\alpha, \beta \cdot \Delta_{\ell^*}(w_t)\} \tag{15}$$

*for all $t$ and $\eta \leq \eta_0$. Then an algorithm based on the EXO approach achieves the following regret bounds simultaneously: (i) In adversarial environments, we have $R_T = O\left(\sqrt{\alpha\nu T \log T} + \frac{\nu \log T}{\eta_0}\right)$.*

*(ii) In environments satisfying (2), we have $R_T(a^*) = O\left(\left(\beta + \frac{1}{\eta_0}\right)\nu \log T + \sqrt{C\beta\nu \log T}\right)$.*

### 5.1 Connection to the SCRiBLe algorithm and variants of it

Abernethy et al. [2008, 2012] have proposed a sampling scheme $w_t \mapsto p_t$ supported on the Dikin ellipsoid, which leads to $\Lambda_{w_t,\eta_t}(p_t, h_t) = O(\eta_t d^2)$ with an appropriately defined unbiased estimator $h_t$. This algorithm referred to as SCRiBLe hence achieves $R_T = O(\sqrt{d^2 \nu T \log T})$, the adversarial regret bound in Lemma 7 with $\alpha = O(d^2)$. Recent studies by Dann et al. [2023b, Theorem 3] and Ito and Takemura [2023] have shown that a modified sampling scheme based on the Dikin ellipsoid sampling achieves $\Lambda_{w_t,\eta_t}(p_t, h_t) = O\left(\eta_t d^2 \cdot \min\{1, \Delta_{\ell^*}(w_t)/\Delta_{\ell^*,\min}\}\right)$. Consequently, these modified algorithms achieve BOBW regret bounds in Lemma 7 with $\alpha = O(d^2)$ and $\beta = O(d^2/\Delta_{\ell^*,\min})$. These results suggest that the following lemma holds:

**Lemma 8** ([Dann et al., 2023b, Ito and Takemura, 2023]). *If $\psi$ is a self-concordant barrier, (15) holds with $\alpha = O(d^2)$, $\beta = O(d^2/\Delta_{\ell^*,\min})$ and $\eta_0 = O(1/d)$.*

### 5.2 Connection to EXO with continuous exponential weights

The entropic barrier is a $\nu$-self-concordant barrier with $\nu \leq d$ [Chewi, 2021], for which a definition and properties are given, e.g., in [Bubeck and Eldan, 2019]. We can see that mean-oriented EXO with entropic-barrier regularization coincides with the EXO approach with exponential weights. In fact, if we compute $w_t$ using (13) with $\psi$ the entropic barrier, from the definition of the entropic barrier, $w_t$ is equal to the mean of the distribution $q_t$ given by the continuous exponential weights. Given this correspondence, we can see that bounds on $\Lambda_{w_t,\eta}^*$ follow immediately from Corollary 1:

**Lemma 9.** *If $\psi$ is the entropic barrier for $\mathcal{X}$, (15) holds with $\alpha = O(d)$, $\beta = O(d/\Delta_{\ell^*,\min})$ and $\eta_0 = O(1)$.*

It is worth noting that these bounds are $O(1/d)$-times better than the results for Lemma 8 that follow from previous studies.

# 6 Limitations and future work

A limitation of this work is that the regret bounds for (corrupted) stochastic environments require the assumption that the optimal arm $a^* \in \arg\min_{a \in \mathcal{A}} \langle \ell^*, a \rangle$ is unique. While this assumption is common in analyses of best-of-both-worlds algorithms based on the self-bounding technique [Zimmert and Seldin, 2021, Dann et al., 2023b], a limited number of studies on the multi-armed bandit problem [Ito, 2021, Jin et al., 2023] have removed this uniqueness assumption by careful analyses of regret bounds for follow-the-regularized-leader methods. As these analyses rely on structures specific to the multi-armed bandit problems, extending them to other problem settings does not seem trivial. Extending this analysis technique to linear bandits and removing the uniqueness assumptions will be an important future challenge.

Another future direction is to work toward a tighter regret bound of $O(c^* \log T)$ in stochastic environments while satisfying BOBW regret guarantee, where $c^*$ is an instance-dependent quantity that characterizes the optimal asymptotic bound (see, e.g., [Lattimore and Szepesvari, 2017, Corollary 2]). As $c^* \leq O(d/\Delta_{\min})$ [Lee et al., 2021, Lemma 16.] and the gap between these two quantities can be arbitrarily large [Lattimore and Szepesvari, 2017, Example 4], our bounds depending on $1/\Delta_{\min}$ can be far from tight. In the multi-armed bandit problem, a special case of linear bandits, the quantity $c^*$ can be expressed as $c^* = \sum_{a \in \mathcal{A}: \Delta_{\ell^*}(a) > 0} \frac{2}{\Delta_{\ell^*}(a)}$, and known FTRL-type BOBW algorithms with $O(c^* \log T)$-regret bounds employs the Tsallis entropy with $\Theta(1/\sqrt{t})$-learning rates: $\psi_t(w) = -\sqrt{t} \sum_{i=1}^{d} \sqrt{w_i}$ [Zimmert and Seldin, 2021] or logarithmic barriers with entry-wise adaptive learning rates: $\psi_t(w) = -\sum_{i=1}^{d} \frac{1}{\eta_{ti}} \log w_i$ [Ito, 2021, Ito et al., 2022a]. The latter would be more closely related to our results in this paper as logarithmic barriers are examples of self-concordant barriers. A major difference between the approach in this paper and the one by Ito [2021] is that the former only considers regularizer functions expressed as a scalar multiple of a fixed function, while the latter even changes the *shape* of regularizer functions for each round. Such a more flexible adaptive regularization may be necessary for pursuing tighter regret bounds. Another promising approach is to extend the Tsallis-entropy approach to linear bandits while the connection between the Tsallis-INF [Zimmert and Seldin, 2021] and this paper may be somewhat tenuous as the Tsallis entropy is not a self-concordant barrier.

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

# A   Notes on notation and assumption

For any two real vectors $x = (x_1, \ldots, x_d) \in \mathbb{R}^d$ and $y = (y_1, \ldots, y_d) \in \mathbb{R}^d$, $\langle x, y \rangle$ denotes the the inner product of $x$ and $y$, i.e., $\langle x, y \rangle = \sum_{i=1}^d x_i y_i$. For any real vector $x = (x_1, \ldots, x_d) \in \mathbb{R}^d$, let $\|x\|_1$ denote the $L_1$-norm of $x$, i.e., $\|x\|_1 = \sum_{i=1}^d |x_i|$. For any real symmetric matrix $X$, let $\mathrm{tr}(X)$ denote the trace of $X$. For any two real symmetric matrices $X$ and $Y$, we denote $X \preceq Y$ (or $Y \succeq X$) if and only if $Y - X$ is a positive semi-definite matrix. For any two real symmetric matrices $X$ and $Y$, $X \bullet Y$ denotes the Frobenius inner product of $X$ and $Y$, i.e., $X \bullet Y = \mathrm{tr}(AB)$.

Without loss of generality, we may assume that $\mathcal{A}$ spans all vectors in $\mathbb{R}^d$. Indeed, if not, we can make $\mathcal{A}$ make full-dimensional by ignoring redundant dimensions. Under this assumption, distributions $q$ given by exponential weight methods have non-singular variance-covariance matrices $V(q)$ (and thus also $H(q)$). This fact is used in some places in this paper without notice.

# B   Omitted proofs

## B.1   Lemmas on log-concave distributions

**Lemma 10** (Theorem 5.2 in [Lovász and Vempala, 2007]). *If $X \in \mathbb{R}^n$ is a random variable following a log-concave distribution, for any constant matrix $M \in \mathbb{R}^{m \times n}$, $MX$ follows a log-concave distribution as well.*

**Lemma 11** (Lemmas 5.4 and 5.5 (a) in [Lovász and Vempala, 2007]). *If $p$ is a one-dimensional log-concave distribution, we have*

$$\Pr_{X \sim \alpha}[X \leq \mu(p)] \geq \frac{1}{\mathrm{e}}. \tag{16}$$

*Further, if $V(p) = 1$, we have*

$$\Pr_{X \sim p}[a \leq X \leq b] \leq |b - a| \tag{17}$$

*for any $a, b \in \mathbb{R}$.*

**Lemma 12.** *For any one dimensional log-concave distribution $p$ over a one-dimensional segment $[L, R] \subseteq \mathbb{R}$, we have*

$$V(p) \leq \mathrm{e}^2 \min \left\{ (\mu(p) - L)^2, (R - \mu(p))^2 \right\}. \tag{18}$$

*Proof.* From Lemma 11, we have

$$\frac{1}{\mathrm{e}} \leq \Pr_{X \sim p}[X \leq \mu(p)] = \Pr_{X \sim p}\left[\frac{L}{\sqrt{V(p)}} \leq \frac{X}{\sqrt{V(p)}} \leq \frac{\mu(p)}{\sqrt{V(p)}}\right] \leq \frac{\mu(p) - L}{\sqrt{V(p)}}, \tag{19}$$

where the first and the second inequalities follow from (16) and (17), respectively. We hence have $V(p) \leq \mathrm{e}^2(\mu(p) - L)^2$. By considering $-X$ instead of $X$, we can show $V(p) \leq \mathrm{e}^2(R - \mu(p))^2$ in a similar way. $\square$

**Lemma 13.** *Suppose $\phi$ is defined as in (7). If $y$ follows a log-concave distribution over $\mathbb{R}$ and if $s = \sqrt{\mathbf{E}[y^2]} \leq \sqrt{1/2}$, we have*

$$\mathbf{E}[\phi(y)] \leq s^2 + \frac{\exp(3 - s^{-1})}{1 - \exp(1 - s^{-1})} = O\left(s^2\right). \tag{20}$$

*Proof.* See, e.g., the proof of [Ito et al., 2020, Lemma 5]. $\square$

## B.2 Proof of Lemma 1

*Proof.* From the definition of the regret, we have

$$\mathop{\mathbf{E}}_{a^*\sim p^*}[R_T(a^*)] = \mathop{\mathbf{E}}_{a^*\sim p^*}\left[\sum_{t=1}^{T}\mathop{\mathbf{E}}_{a\sim p_t}[\langle \ell_t, a-a^*\rangle]\right]$$

$$= \mathop{\mathbf{E}}_{a^*\sim p^*}\left[\sum_{t=1}^{T}\left(\mathop{\mathbf{E}}_{x\sim q_t}[\langle \ell_t, x-a^*\rangle] + \langle \ell_t, \mu(q_t)-\mu(p_t)\rangle\right)\right]. \tag{21}$$

From the definition of $\mathrm{bias}$ given in (6), the part of $\mathbf{E}_{x\sim q_t}[\langle \ell_t, x-a^*\rangle]$ can be bounded as follows:

$$\mathop{\mathbf{E}}_{x\sim q_t}[\langle \ell_t, x-a^*\rangle]$$

$$= \mathop{\mathbf{E}}_{a_t,f_t}\left[\mathop{\mathbf{E}}_{x\sim q_t}[\langle \ell_t, x-a^*\rangle - g_t(a^*;a_t,f_t) + g_t(x;a_t,f_t)] + \mathop{\mathbf{E}}_{x\sim q_t}[g_t(x;a_t,f_t)-g_t(a^*;a_t,f_t)]\right]$$

$$\leq \mathrm{bias}(g_t,p_t,q_t,\ell_t) + \mathop{\mathbf{E}}_{a_t,f_t}\left[\mathop{\mathbf{E}}_{x\sim q_t}[g_t(x;a_t,f_t)-g_t(a^*;a_t,f_t)]\right]. \tag{22}$$

From this and (21), we have

$$\mathop{\mathbf{E}}_{a^*\sim p^*}[R_T(a^*)] \leq \mathbf{E}\left[\sum_{t=1}^{T}\left(\langle \ell_t,\mu(q_t)-\mu(p_t)\rangle + \mathrm{bias}(g_t,p_t,q_t,\ell_t)\right)\right]$$

$$+ \mathop{\mathbf{E}}_{a^*\sim p^*}\left[\mathbf{E}\left[\sum_{t=1}^{T}\mathop{\mathbf{E}}_{x\sim q_t}[g_t(x;a_t,f_t)-g_t(a^*;a_t,f_t)]\right]\right]. \tag{23}$$

By a standard analysis of the exponential weight method, we have

$$\mathop{\mathbf{E}}_{a^*\sim p^*}\left[\sum_{t=1}^{T}\mathop{\mathbf{E}}_{x\sim q_t}[g_t(x;a_t,f_t)-g_t(a^*;a_t,f_t)]\right] \leq \sum_{t=1}^{T}\frac{1}{\eta_t}\mathrm{stab}(\eta_t g_t(\cdot;a_t,f_t),q_t) + \frac{1}{\eta_{T+1}}D_{\mathrm{KL}}(p^*,q_0). \tag{24}$$

To show (24), we denote $G_t(p) = \mathbf{E}_{a\sim p}[g_t(a;a_t,f_t)]$ in the following for notational simplicity. From (5), we have

$$\mathop{\mathbf{E}}_{a^*\sim p^*}\left[\sum_{t=1}^{T}g_t(a^*;a_t,f_t)\right] + \frac{1}{\eta_{T+1}}D_{\mathrm{KL}}(p^*,q_0)$$

$$= \sum_{t=1}^{T}G_t(p^*) + \frac{1}{\eta_{T+1}}D_{\mathrm{KL}}(p^*,q_0)$$

$$\geq \sum_{t=1}^{T}G_t(q_{T+1}) + \frac{1}{\eta_{T+1}}D_{\mathrm{KL}}(q_{T+1},q_0)$$

$$\geq \sum_{t=1}^{T-1}G_t(q_{T+1}) + \frac{1}{\eta_T}D_{\mathrm{KL}}(q_{T+1},q_0) + G_T(q_{T+1})$$

$$= \sum_{t=1}^{T-1}G_t(q_T) + \frac{1}{\eta_T}D_{\mathrm{KL}}(q_T,q_0) + \frac{1}{\eta_T}D_{\mathrm{KL}}(q_{T+1},q_T) + G_T(q_{T+1})$$

$$\geq \cdots \geq \sum_{t=1}^{T}\left(\frac{1}{\eta_t}D_{\mathrm{KL}}(q_{t+1},q_t) + G_t(q_{t+1})\right), \tag{25}$$

where the first inequality follows from (5) and the second inequality follows from $\eta_{T+1} \leq \eta_T$ and the fact that the values of the KL divergence are nonnegative. The second equality follows from the definition of $q_T(x)$; As $q_T(x)$ can be expressed as $q_T(x) = \kappa_T \exp(-\eta_T \tilde{g}_T(x))q_0(x)$ for some

$\kappa_T > 0$, where we denote $\tilde{g}_T(x) = \sum_{s=1}^{T-1} g_s(x; a_s, f_s)$, we have

$$\sum_{t=1}^{T-1} G_t(q_T) + \frac{1}{\eta_T} D_{\mathrm{KL}}(q_T, q_0) = \mathop{\mathbf{E}}_{a \sim q_T} \left[ \tilde{g}_T(a) + \frac{1}{\eta_T} \ln \frac{q_T(a)}{q_0(a)} \right]$$

$$= \mathop{\mathbf{E}}_{a \sim q_T} \left[ \tilde{g}_T(a) + \frac{1}{\eta_T} \left( \ln \kappa_T - \eta_T \tilde{g}_T(x) \right) \right] = \frac{\ln \kappa_T}{\eta_T} \qquad (26)$$

and

$$\sum_{t=1}^{T-1} G_t(q_{T+1}) + \frac{1}{\eta_T} D_{\mathrm{KL}}(q_{T+1}, q_0) = \mathop{\mathbf{E}}_{a \sim q_{T+1}} \left[ \tilde{g}_T(a) + \frac{1}{\eta_T} \ln \frac{q_{T+1}(a)}{q_0(a)} \right]$$

$$= \mathop{\mathbf{E}}_{a \sim q_{T+1}} \left[ \tilde{g}_T(a) + \frac{1}{\eta_T} \ln \frac{q_{T+1}(a)}{q_T(a)} + \frac{1}{\eta_T} \ln \frac{q_T(a)}{q_0(a)} \right]$$

$$= \mathop{\mathbf{E}}_{a \sim q_{T+1}} \left[ \tilde{g}_T(a) + \frac{1}{\eta_T} \ln \frac{q_{T+1}(a)}{q_T(a)} + \frac{1}{\eta_T} \left( \ln \kappa_T - \eta_T \tilde{g}_T(x) \right) \right]$$

$$= \frac{\ln \kappa_T}{\eta_T} + \frac{1}{\eta_T} \mathop{\mathbf{E}}_{a \sim q_{T+1}} \left[ \ln \frac{q_{T+1}(a)}{q_T(a)} \right] = \frac{\ln \kappa_T}{\eta_T} + D_{\mathrm{KL}}(q_{T+1}, q_T). \qquad (27)$$

These equalities together imply that the second equality in (25). The last inequality of (25) can be obtained by applying similar transformations repeatedly. From (25), we have

$$[\text{LHS of (24)}] = \sum_{t=1}^{T} \left( G_t(q_t) - G_t(p^*) \right)$$

$$\le \sum_{t=1}^{T} \left( G_t(q_t) - G_t(q_{t+1}) - \frac{1}{\eta_t} D_{\mathrm{KL}}(q_{t+1}, q_t) \right) + \frac{1}{\eta_{T+1}} D_{\mathrm{KL}}(p^*, q_0)$$

$$\le \sum_{t=1}^{T} \frac{1}{\eta_t} \mathrm{stab}(\eta_t g_t(\cdot; a_t, f_t), q_t) + \frac{1}{\eta_{T+1}} D_{\mathrm{KL}}(p^*, q_0), \qquad (28)$$

which implies that (24) holds. By combining (23) and (24), we obtain the bound of (6). $\qquad \square$

### B.3 Proof of Theorem 1

*Proof.* For any $a^* \in \mathcal{X}$, let $p^*$ be the uniform distribution over $\{ (1 - \frac{1}{T}) a^* + \frac{1}{T} x \mid x \in \mathcal{X} \}$. We then have $R_T(a^*) \le \mathbf{E}_{a \sim p^*}[R_T(a)] + 1$ and $D_{\mathrm{KL}}(p^*, q_1) \le d \log T$. Define $\eta_t$ by $\eta_t = \min \left\{ \eta_0, \sqrt{d \log T} / \sqrt{\alpha + \sum_{s=1}^{t-1} \eta_s^{-1} \Lambda_{q_s, \eta_s}^*} \right\}$. Suppose that $p_t$ and $g_t$ are chosen so that $\Lambda_{q_t, \eta_t}(p_t, g_t) = \Lambda_{q_t, \eta_t}^*$. As we have $\eta_t \le \eta_0$, from the assumption, we have $\eta_t^{-1} \Lambda_{q_t, \eta_t} \le \alpha$. Hence, from (8), we have

$$R_T \le \mathbf{E} \left[ \sum_{t=1}^{T} \Lambda_{q_t, \eta_t}^* + \frac{d \log T}{\eta_{T+1}} \right] + 1 \le \mathbf{E} \left[ \sum_{t=1}^{T} \frac{\sqrt{d \log T} \eta_t^{-1} \Lambda_{q_t, \eta_t}^*}{\sqrt{\alpha + \sum_{s=1}^{t-1} \eta_s^{-1} \Lambda_{q_s, \eta_s}^*}} + \frac{d \log T}{\eta_{T+1}} \right] + 1$$

$$\le 3 \mathbf{E} \left[ \sqrt{d \log T \left( \alpha + \sum_{t=1}^{T} \eta_t^{-1} \Lambda_{q_t, \eta_t}^* \right)} \right] + \frac{d \log T}{\eta_0} + 1$$

$$\le 3 \mathbf{E} \left[ \sqrt{d \log T \left( \alpha + \min \left\{ \alpha T, \beta \sum_{t=1}^{T} \Delta_{\ell^*}(\mu(q_t)) \right\} \right)} \right] + \frac{d \log T}{\eta_0} + 1 =: R_T', \qquad (29)$$

which implies the regret bound of $R_T = O\left( \sqrt{\alpha d T \log T} + \frac{d \log T}{\eta_0} \right)$ that holds in adversarial environments. We next show the improved regret bounds for stochastically constrained adversarial environments. From the definition of $\Lambda$, we have

$$\Lambda_{q_t, \eta_t}^* = \Lambda_{q_t, \eta_t}(p_t, g_t) \ge \sup_{\ell \in \mathcal{L}} \left\{ \langle \ell, \mu(p_t) - \mu(q_t) \rangle \right\} \ge \langle \ell^*, \mu(q_t) - \mu(p_t) \rangle, \qquad (30)$$

where the last inequality holds since $-\ell^* \in \mathcal{L}$ for any $\ell^* \in \mathcal{L}$, which follows from the definition of $\mathcal{L}$. We then, hence, have

$$R_T'' := \mathbf{E}\left[\sum_{t=1}^T \Delta_{\ell^*}(\mu(q_t))\right] = \mathbf{E}\left[\sum_{t=1}^T \left(\Delta_{\ell^*}(\mu(p_t)) + \langle \ell^*, \mu(q_t) - \mu(p_t)\rangle\right)\right]$$

$$\le \mathbf{E}\left[\sum_{t=1}^T \Delta_{\ell^*}(\mu(p_t))\right] + \mathbf{E}\left[\sum_{t=1}^T \Lambda_{q_t,\eta_t}^*\right] \le R_T + C + R_T' \le 2R_T' + C. \qquad (31)$$

From this, we have

$$R_T' \le 3\,\mathbf{E}\left[\sqrt{d\log T\,(\alpha + \beta R_T'')}\right] + \frac{d\log T}{\eta_0} + 1 \le 3\,\mathbf{E}\left[\sqrt{d\log T\,(\alpha + \beta(2R_T' + C))}\right] + \frac{d\log T}{\eta_0} + 1$$

which implies that $R_T = O\left(\left(\beta + \frac{1}{\eta_0}\right)d\log T + \sqrt{C\beta d\log T}\right)$ holds. $\qquad\square$

## B.4 Proof of Lemma 2

*Proof.* We first see that $\Lambda_{q,\eta}^* = O(1)$ holds for any $q \in \mathcal{P}(\mathcal{X})$ and $\eta > 0$. In fact, by setting $p = q$ and letting $g \in \mathcal{G}$ to be the zero function ($g(x; a, f) = 0$ for all $x, a$ and $f$), we obtain $\Lambda_{q,\eta}(p, g) = \sup_{\ell \in \mathcal{L}} \mathrm{bias}(g, p, q, \ell) = \sup_{\ell \in \mathcal{L}, x \in \mathcal{X}, a^* \in \mathcal{X}} \langle \ell, x - a^*\rangle \le 2$.

Let us consider the case in which $\eta \le 1/d$ holds. As $\Lambda_{q,\eta}^* = O(1)$ for any $\eta$ and $q$, we have $\Lambda_{q,\eta}^* = O(\eta\omega(q))$ if $\eta\omega(q) = \Omega(1)$. In the following, we assume $\eta\omega(q) \le 1/8$. Let $\tilde{p} \in \mathcal{P}(\mathcal{X})$ be a distribution such that $\mu(\tilde{p}) = \mu(q)$ and $w(q) = H(\tilde{p})^{-1} \bullet V(q)$. Let $\pi_0 \in \mathcal{P}(\mathcal{X})$ be a distribution such that

$$a^\top H(\pi_0)^{-1} a \le d \qquad (32)$$

holds for any $a \in \mathcal{A}$. Note that such a distribution always exists (see, e.g., [Lattimore and Szepesvári, 2020a, Theorem 21.1]). Define $p$ by

$$p = (1 - \gamma)\tilde{p} + \gamma\pi_0 \qquad (33)$$

with $\gamma = 4\eta\omega(q) \le 1/2$. With this $p$, we define $g$ by (4). Then, as we have $\mathrm{bias}(g, p, q, \ell) = 0$, we have

$$\Lambda_{q,\eta}(p, g) \le \sup_{\ell \in \mathcal{A}} \{\langle \ell, \mu(p) - \mu(q)\rangle\} + \frac{1}{\eta} \sup_{f \in [-1,1]} \mathbf{E}_{a \sim p}[\mathrm{stab}(\eta g(\cdot; a, f), q)]. \qquad (34)$$

As the definition of $p$ implies $\mu(p) = (1 - \gamma)\mu(\tilde{p}) + \gamma\mu(\pi_0) = (1 - \gamma)\mu(q) + \gamma\mu(\pi_0)$, it holds for any $\ell \in \mathcal{L}$ that

$$\langle \ell, \mu(p) - \mu(q)\rangle = \langle \ell, (1 - \gamma)\mu(q) + \gamma\mu(\pi_0) - \mu(q)\rangle = \gamma\langle \ell, \gamma\mu(\pi_0) - \mu(q)\rangle \le 2\gamma. \qquad (35)$$

Further, from (7) and (4), we have

$$\mathrm{stab}(\eta g(\cdot; a, f), q) \le \mathbf{E}_{x \sim q}[\phi(\eta g(x; a, f) - \eta g(\mu(q); a, f))]$$

$$= \mathbf{E}_{x \sim q}\left[\phi\left(\eta f \cdot \langle (H(p))^{-1} a, x - \mu(q)\rangle\right)\right] \qquad (36)$$

for any fixed $a \in \mathcal{A}$ and $f \in [-1, 1]$. Let $r$ represent the distribution of $f \cdot \langle (H(p))^{-1} a, x - \mu(q)\rangle$ for $x \sim q$ with any fixed $a \in \mathcal{A}$ and $f \in [-1, 1]$. Then, as $q$ is a log-concave distribution, from Lemma 10, $r$ is a log-concave distribution as well. Further, we have

$$H(r) = \mathbf{E}_{y \sim r}[y^2] = f^2\,\mathbf{E}_{x \sim q}\left[\left(\langle (H(p))^{-1} a, x - \mu(q)\rangle\right)^2\right]$$

$$= f^2 a^\top (H(p))^{-1}\,\mathbf{E}_{x \sim q}\left[(x - \mu(q))(x - \mu(q))^\top\right](H(p))^{-1} a$$

$$= f^2 a^\top (H(p))^{-1} V(q)(H(p))^{-1} a \le a^\top (H(p))^{-1} V(q)(H(p))^{-1} a. \qquad (37)$$

Since we have

$$\mathrm{tr}(H(p)^{-1/2} V(q) H(p)^{-1/2}) = V(q) \bullet H(p)^{-1} = V(q) \bullet ((1 - \gamma)H(\tilde{p}) + \gamma H(\pi_0))^{-1}$$

$$\le V(q) \bullet ((1 - \gamma)H(\tilde{p}))^{-1} = \frac{1}{1 - \gamma}\omega(q), \qquad (38)$$

we have

$$H(p)^{-1/2}V(q)H(p)^{-1/2} \preceq \frac{\omega(q)}{1-\gamma}I_d \tag{39}$$

where $I_d$ represents the identity matrix of size $d$. Hence, we have

$$
\begin{aligned}
H(r) &\leq a^\top (H(p))^{-1/2}\left(\frac{\omega(q)}{1-\gamma}I_d\right)(H(p))^{-1/2}a \leq \frac{\omega(q)}{1-\gamma}a^\top (H(p))^{-1}a \\
&= \frac{\omega(q)}{1-\gamma}a^\top ((1-\gamma)H(\tilde{p})+\gamma H(\pi_0))^{-1}a \leq \frac{\omega(q)}{1-\gamma}a^\top (\gamma H(\pi_0))^{-1}a \\
&= \frac{\omega(q)}{(1-\gamma)\gamma}a^\top H(\pi_0)^{-1}a \leq \frac{\omega(q)d}{(1-\gamma)\gamma} \leq 2\frac{\omega(q)d}{\gamma} = \frac{d}{2\eta}.
\end{aligned}
\tag{40}
$$

This implies that $\eta^2 H(r) \leq d\eta/2 \leq 1/2$. Hence, from Lemma 13, we have

$$\mathrm{stab}(\eta g(\cdot; a, f), q) \leq \underset{y\sim r}{\mathbf{E}}[\phi(\eta y)] = O\left(\eta^2 H(r)\right) = O\left(\eta^2 a^\top (H(p))^{-1}V(q)(H(p))^{-1}a\right). \tag{41}$$

Hence, we have

$$
\begin{aligned}
\sup_{f\in[-1,1]}\underset{a\sim p}{\mathbf{E}}[\mathrm{stab}(\eta g(\cdot; a, f), q)] &= O\left(\eta^2\underset{a\sim p}{\mathbf{E}}\left[a^\top (H(p))^{-1}V(q)(H(p))^{-1}a\right]\right) \\
&= O\left(\eta^2\mathrm{tr}\left(\underset{a\sim p}{\mathbf{E}}\left[aa^\top\right](H(p))^{-1}V(q)(H(p))^{-1}\right)\right) = O\left(\eta^2 V(q)\bullet (H(p))^{-1}\right) \\
&\leq O\left(\frac{\eta^2}{1-\gamma}V(q)\bullet (H(\tilde{p}))^{-1}\right) = O\left(\frac{\eta^2\omega(q)}{1-\gamma}\right) = O\left(\eta^2\omega(q)\right).
\end{aligned}
\tag{42}
$$

Combining this with (34) and (35), we obtain

$$\Lambda_{q,\eta}(p,g) \leq 2\gamma + \frac{1}{\eta}\sup_{f\in[-1,1]}\underset{a\sim p}{\mathbf{E}}[\mathrm{stab}(\eta g(\cdot; a, f), q)] = O\left(\gamma + \eta\omega(q)\right) = O\left(\eta\omega(q)\right), \tag{43}$$

which implies that $\Lambda^*_{q,\eta} = O(\eta\omega(q))$.

We next consider the case in which $\eta \leq 1$ holds. As we have $\Lambda^*_{q,\eta} = O(\eta\omega'(q))$ if $\eta\omega'(q) > 1/8$ (since $\Lambda^*_{q,\eta} = O(1)$ for any $q$ and $\eta$), we assume $\eta\omega'(q) \leq 1/8$. We set $p$ by (33) where $\tilde{p}$ is the minimizer in the right-hand side of (12) and $\gamma = 4\eta\omega'(q) \leq 1/2$. We set $g$ by (4). Then, (34) and (35) can be shown in a similar way to in the first half of this proof. As we have

$$dV(q) \preceq \omega'(q)H(\tilde{p}) \preceq \frac{\omega'(q)}{1-\gamma}H(p), \tag{44}$$

we have

$$H(p)^{-1/2}V(q)H(p)^{-1/2} \preceq \frac{\omega'(q)}{(1-\gamma)d}I_d. \tag{45}$$

From this, if we define $r$ in a similar way to in the first half of this proof, we have $H(r) \leq 1/(2\eta)$. We hence have $\eta^2 H(r) \leq \eta/2 \leq 1/2$ under the assumption of $\eta \leq 1$. We can then apply Lemma 13 to obtain the following:

$$
\begin{aligned}
\sup_{f\in[-1,1]}\underset{a\sim p}{\mathbf{E}}[\mathrm{stab}(\eta g(\cdot; a, f), q)] &\leq O\left(\frac{\eta^2}{1-\gamma}V(q)\bullet (H(\tilde{p}))^{-1}\right) \\
&\leq O\left(\frac{\eta^2}{1-\gamma}\frac{\omega'(q)}{d}H(\tilde{p})\bullet (H(\tilde{p}))^{-1}\right) = O\left(\frac{\eta^2\omega'(q)}{1-\gamma}\right) = O\left(\eta^2\omega'(q)\right),
\end{aligned}
\tag{46}
$$

where the first inequality follows from a similar argument as in (42). Combining this with (34), (35), and $\gamma = 4\eta\omega'(q)$, we obtain $\Lambda_{q,\eta}(p,g) = O(\eta\omega'(q))$, which implies $\Lambda^*_{q,\eta} = O(\eta\omega'(q))$. $\qquad\square$

## B.5 Proof of Lemma 3

*Proof of Lemma 3.* We will show that

$$\text{there exists} \quad p \in \mathcal{P}(\mathcal{X}) \quad \text{such that} \quad \mu(p) = \mu(q) \quad \text{and} \quad V(q) \preceq z \cdot V(p) \tag{47}$$

with $z = O(1, \Delta_{\ell^*}(\mu(q))/\Delta_{\ell^*,\min})$. As we have $H(p) = V(p) + \mu(p)\mu(p)^\top \succeq V(p)$, (47) leads to $d \cdot V(q) \preceq dz \cdot V(p) \preceq dz \cdot H(p)$, which implies $\omega'(q) \leq dz$. Hence, to prove the lemma, it is sufficient to show that (47) holds with $z = O(1, \Delta_{\ell^*}(\mu(q))/\Delta_{\ell^*,\min})$. By considering the case of $p = q$, we can easily see that (47) holds with $z = O(1)$. for any $q$. In the following, we show that (47) holds with $z = O\left(\frac{\Delta_{\ell^*}(\mu(q))}{\Delta_{\ell^*,\min}}\right)$.

Without loss of generality, we may assume $\mu(q) = 0$, via a variable transformation $x \leftarrow x - \mu(q)$. In fact, such a transformation preserves $V(q)$ and $V(p)$. For any $x \in \mathcal{X}$, there exists $\xi_x = (\xi_x(a))_{a \in \mathcal{A}} \in \mathcal{P}(\mathcal{A})$ such that $\mu(\xi_x) = \sum_{a \in \mathcal{A}} \xi_x(a) = x$. For any $(\xi_x)_{x \in \mathcal{X}}$, define $p \in \mathcal{P}(\mathcal{A})$ by $p(a) = \mathbf{E}_{x \sim q}[\xi_x(a)]$. We then have $\mu(p) = \mathbf{E}_{a \sim p}[x] = \mathbf{E}_{x \sim q}[\mathbf{E}_{a \sim \xi_x}[a]] = \mathbf{E}_{x \sim q}[x] = \mu(q) = 0$. Fix an arbitrary non-zero vector $u \in \mathbb{R}^d \setminus \{0\}$ and an arbitrary $a^* \in \mathcal{A}$. Denote $\varepsilon = 1 - \Pr_{a \sim p}[a = a^*]$.

Set $v = u/\sqrt{u^\top V(q)u}$. Let $p_v$ and $q_v$ denote the distributions of $\langle v, a \rangle$ for $a \sim p$ and $\langle v, x \rangle$ for $x \sim q$, respectively. We then have $V(q_v) = v^\top V(q)v = u^\top V(q)u/(u^\top V(q)u) = 1$. Without loss of generality, we assume that $\langle v, a^* \rangle \geq \langle v, \mu(q) \rangle = \mu(q_v) = 0$. (If not, we redefine $v = -u/\sqrt{u^\top V(q)u}$.) Since $q$ is a log-concave distribution, from Lemma 10, $q_v$ is a log-concave distribution as well. Hence, from Lemma 11, we have

$$\Pr_{X \sim q_v}\left[X \leq -\frac{1}{2e}\right] = \Pr_{X \sim q_v}[X \leq 0] - \Pr_{X \sim q_v}\left[-\frac{1}{2e} < X \leq 0\right] \geq \frac{1}{e} - \frac{1}{2e} = \frac{1}{2e}, \tag{48}$$

where the inequality follows from (16) and (17).

We hence have

$$V(p_v) = \mathbf{E}_{a \sim p}[(\langle v, a \rangle)^2] = \mathbf{E}_{x \sim q}\left[\mathbf{E}_{a \sim \xi_x}[(\langle v, a \rangle)^2]\right]$$

$$\geq \mathbf{E}_{x \sim q}\left[\mathbf{E}_{a \sim \xi_x}[(\langle v, a \rangle)^2] \mid \langle v, x \rangle \leq -\frac{1}{2e}\right] \cdot \Pr_{X \sim q_v}\left[X \leq -\frac{1}{2e}\right]$$

$$\geq \frac{1}{2e} \cdot \mathbf{E}_{x \sim q}\left[\mathbf{E}_{a \sim \xi_x}[(\langle v, a \rangle)^2] \mid \langle v, x \rangle \leq -\frac{1}{2e}\right], \tag{49}$$

where the last inequality follows from (48). Suppose $\langle v, x \rangle \leq -\frac{1}{2e}$. Since $\mu(\xi_x) = x$, we have

$$-\frac{1}{2e} \geq \langle v, x \rangle = \xi_x(a^*)\langle v, a^* \rangle + \sum_{a \in \mathcal{A} \setminus \{a^*\}} \xi_x(a)\langle v, a \rangle \geq \sum_{a \in \mathcal{A} \setminus \{a^*\}} \xi_x(a)\langle v, a \rangle. \tag{50}$$

We hence have

$$\mathbf{E}_{a \sim \xi_x}[(\langle v, a \rangle)^2] \geq \sum_{a \in \mathcal{A} \setminus \{a^*\}} \xi_x(a)(\langle v, a \rangle)^2 = (1 - \xi_x(a^*))\frac{\sum_{a \in \mathcal{A} \setminus \{a^*\}} \xi_x(a)(\langle v, a \rangle)^2}{\sum_{a \in \mathcal{A} \setminus \{a^*\}} \xi_x(a)}$$

$$\geq (1 - \xi_x(a^*))\left(\frac{\sum_{a \in \mathcal{A} \setminus \{a^*\}} \xi_x(a)\langle v, a \rangle}{\sum_{a \in \mathcal{A} \setminus \{a^*\}} \xi_x(a)}\right)^2 = \frac{\left(\sum_{a \in \mathcal{A} \setminus \{a^*\}} \xi_x(a)\langle v, a \rangle\right)^2}{1 - \xi_x(a^*)} \geq \frac{1}{4e^2(1 - \xi_x(a^*))},$$

where the second inequality comes from Jensen's inequality, and the last inequality follows from (50). From this and (49), we have

$$V(p_v) \geq \frac{1}{8e^3} \mathbf{E}_{x \sim q}\left[\frac{1}{1 - \xi_x(a^*)} \mid \langle v, x \rangle \leq -\frac{1}{2e}\right] \geq \frac{1}{8e^3} \cdot \frac{1}{\mathbf{E}_{x \sim q}\left[1 - \xi_x(a^*) \mid \langle v, x \rangle \leq -\frac{1}{2e}\right]}. \tag{51}$$

We further have

$$\varepsilon = \mathbf{E}_{x \sim q}[1 - \xi_x(a^*)] \geq \mathbf{E}_{x \sim q}\left[1 - \xi_x(a^*) \mid \langle v, x \rangle \leq -\frac{1}{2e}\right] \cdot \Pr_{x \sim q}\left[\langle v, x \rangle \leq -\frac{1}{2e}\right]$$

$$\geq \frac{1}{2e} \mathbf{E}_{x \sim q}\left[1 - \xi_x(a^*) \mid \langle v, x \rangle \leq -\frac{1}{2e}\right], \tag{52}$$

where the last inequality follows from (48). Combining this with (51), we obtain

$$V(p_v) \geq \frac{1}{16\mathrm{e}^4\varepsilon}. \tag{53}$$

As it follows from the definition of $v$ that

$$V(p_v) = v^\top V(p)v = \frac{u^\top V(p)u}{u^\top V(q)u}, \tag{54}$$

we have

$$16\mathrm{e}^4\varepsilon \cdot u^\top V(p)u \geq u^\top V(q)u. \tag{55}$$

Since this holds for all $u \in \mathbb{R}^d \setminus \{0\}$, we have

$$16\mathrm{e}^4\varepsilon \cdot V(p) \succeq V(q) \tag{56}$$

Letting $a^* \in \arg\min_{a \in \mathcal{A}} \langle \ell^*, a \rangle$, we obtain

$$\Delta_{\ell^*}(\mu(q)) = \Delta_{\ell^*}(\mu(p)) = \sum_{a \in \mathcal{A}} p(a)\Delta_{\ell^*}(a)$$

$$\geq \sum_{a \in \mathcal{A} \setminus \{a^*\}} p(a)\Delta_{\ell^*,\min} = \Delta_{\ell^*,\min} \cdot (1 - p(a^*)) = \Delta_{\ell^*,\min} \cdot \varepsilon \tag{57}$$

Combining this with (56), we obtain

$$16\mathrm{e}^4 \frac{\Delta_{\ell^*}(\mu(q))}{\Delta_{\ell^*,\min}} \cdot V(p) \succeq V(q), \tag{58}$$

which completes the proof.

## B.6 Proof of Lemma 4

*Proof.* Let $q$ be a log-concave distribution over $\mathcal{X} = \{x \in [0,1]^d \mid \|x\|_1 = 1\}$. If $x = (x_1, \ldots, x_d) \sim q$, from Lemma 10, $x_i$ for each $i$ also follows a log-concave distribution over $[0,1]$. Hence, from Lemma 12, the variance of $x_i$ ( i.e., the $(i,i)$ entry of $V(q)$ ) is bounded as

$$[V(q)]_{i,i} \leq \mathrm{e}^2 \min\{\mu_i(q)^2, (1-\mu_i(q))^2\} \leq 4\mathrm{e}^2\mu_i(q)^2(1-\mu_i(q))^2 \tag{59}$$

Consider the distribution $p \in \mathcal{P}(\mathcal{A})$ such that $\Pr_{a \sim p}[a = e_i] = \mu(q)$. We then have $\mu_i(p) = \mu_i(q)$ and that $H(p)$ is a diagonal matrix with diagonal entries $(\mu_i(q))_{i=1}^d$. Hence, from (59), we have

$$V(q) \bullet (H(p))^{-1} = \sum_{i=1}^d [V(q)]_{i,i}(\mu_i(q))^{-1} \leq 4\mathrm{e}^2 \sum_{i=1}^d \mu_i(q)(1-\mu_i(q))^2$$

$$\leq 4\mathrm{e}^2 \sum_{i=1}^d \mu_i(q)(1-\mu_i(q)), \tag{60}$$

which implies that $\omega(q) = O\left(\sum_{i=1}^d \mu_i(q)(1-\mu_i(q))\right)$. Further, as we have $\sum_{i=1}^d \mu_i(q) = 1$, it holds for any $i^* \in \{1, \ldots, d\}$ that

$$\sum_{i=1}^d \mu_i(q)(1-\mu_i(q)) = \mu_{i^*}(q)(1-\mu_{i^*}(q)) + \sum_{i \neq i^*} \mu_i(q)(1-\mu_i(q))$$

$$\leq 1 - \mu_{i^*}(q) + \sum_{i \neq i^*} \mu_i(q) = 2\left(1 - \mu_{i^*}(q)\right), \tag{61}$$

which complete the proof. □

## B.7 Proof of Corollary 2

*Proof.* From Lemma 4 and the fact that $\sum_{i=1}^d \mu_i(q) = 1$, it is clear that $\omega(q) = O(1)$. Let $\ell^* \in \mathcal{L}$ be an a loss vector such that $e_{i^*} \in \arg\min_{1 \leq i \leq d} \ell_i^*$ is unique. We then have $\Delta_{\ell^*}(\mu(q)) = \sum_{i \neq i^*} \Delta_{\ell^*}(e_i)\mu_i \geq \Delta_{\ell^*,\min}(1-\mu_{i^*}(q))$. Hence, from Lemma 4, we have $\omega(q) = O\left(1 - \mu_{i^*}(q)\right) \leq O\left(\Delta_{\ell^*}(\mu(q))/\Delta_{\ell^*,\min}\right)$. From this and Lemma 2, (10) holds with $\alpha = O(1)$, $\beta = O(\frac{1}{\Delta_{\ell^*,\min}})$, and $\eta_0 = O(1/d)$. □

## B.8 Proof of Lemma 5

*Proof.* For each $j \in \{1, \ldots, m\}$, let $p^{(j)} \in \mathcal{P}(\mathcal{X}^{(j)})$ be such that $\mu(p^{(j)}) = \mu(q^{(j)})$ and $\omega(q^{(j)}) = V(q^{(j)}) \bullet (H(p^{(j)}))^{-1}$. Let $p \in \mathcal{P}(\mathcal{X})$ be the product measure of $p^{(1)}, \ldots, p^{(m)}$. As $p$ is the product measure of $q^{(1)}, \ldots, q^{(m)}$, $V(q) \in \mathbb{R}^{d \times d}$ is a block diagonal matrix with submatrices $V(q^{(1)}), \ldots, V(q^{(m)})$. Similarly, $H(p) \in \mathbb{R}^{d \times d}$ is a block diagonal matrix with submatrices $H(p^{(1)}), \ldots, H(p^{(m)})$. We hence have $V(q) \bullet (H(p))^{-1} = \sum_{j=1}^{m} V(q^{(j)}) \bullet (H(p^{(j)}))^{-1} = \sum_{j=1}^{m} \omega(q^{(j)})$, which implies $\omega(q) \leq \sum_{j=1}^{m} \omega(q^{(j)})$. $\qquad\square$

## B.9 Proof of Corollary 3

*Proof.* Suppose $\mathcal{A}$ is given by $\mathcal{A} = \mathcal{A}_{d_1} \times \cdots \times \mathcal{A}_{d_m}$. For any $\ell = (\ell^{(1)}, \ell^{(2)}, \ldots, \ell^{(m)}) \in \mathcal{L}$ and $x = (x^{(1)}, x^{(2)}, \ldots, x^{(m)}) \in \mathcal{X}$, where $\ell^{(j)} \in \mathbb{R}^{d_j}$ and $x^{(j)} \in \mathcal{X}^{(j)}$ for each $j$, denote $\Delta_{\ell^{(j)}}^{(j)}(x^{(j)}) = \langle \ell^{(j)}, x^{(j)} \rangle - \min_{a^{(j)} \in \mathcal{A}^{(j)}} \langle \ell^{(j)}, a^{(j)} \rangle = \langle \ell^{(j)}, x^{(j)} - a^{*(j)} \rangle$. We then have $\Delta_{\ell^*}(x) = \sum_{j=1}^{m} \Delta_{\ell^{(j)}}^{(j)}(x^{(j)})$. Let $\ell$ be a loss vector such that $a^* = (a^{*(1)}, a^{*(2)}, \ldots, a^{*(m)}) \in \arg\min_{a \in \mathcal{A}} \langle \ell, a \rangle$ is unique. We then have $\Delta_{\ell, \min} = \min_{1 \leq j \leq m} \Delta_{\ell^{(j)}, \min}^{(j)}$.

If $q \in \mathcal{X}$ is a product measure of $q_j \in \mathcal{P}(\mathcal{X}^{(j)})$, from Lemmas 4 and 5, we have

$$
\begin{aligned}
\omega(q) &\leq O\left( \sum_{j=1}^{d} \omega(q^{(j)}) \right) \leq O\left( \sum_{j=1}^{m} \min\left\{ 1, \frac{\Delta_{\ell^{*(j)}}^{(j)}(\mu(q^{(j)}))}{\Delta_{\ell^{*(j)}, \min}} \right\} \right) \\
&\leq O\left( \min\left\{ m, \sum_{j=1}^{m} \frac{\Delta_{\ell^{*(j)}}^{(j)}(\mu(q^{(j)}))}{\Delta_{\ell^{*(j)}, \min}} \right\} \right) \leq O\left( \min\left\{ m, \sum_{j=1}^{m} \frac{\Delta_{\ell^{*(j)}}^{(j)}(\mu(q^{(j)}))}{\Delta_{\ell^*, \min}} \right\} \right) \\
&= O\left( \min\left\{ m, \frac{\Delta_{\ell^*}^{(j)}(\mu(q))}{\Delta_{\ell^*, \min}} \right\} \right).
\end{aligned} \tag{62}
$$

This means that (10) holds with $\alpha = O(m)$ and $\beta = O(\frac{1}{\Delta_{\ell^*, \min}})$.

The case of $\mathcal{A} = \{0, 1\}^d$ can be interpreted as a problem instance in which $\mathcal{A}$ is a direct product of $d$ copies of $\mathcal{A}_2$. Hence, we can see that (10) holds with $\alpha = O(d)$ and $\beta = O(\frac{1}{\Delta_{\ell^*, \min}})$. $\qquad\square$

## B.10 Proof of Lemma 6

*Proof.* From the definition of the regret, we have

$$
\begin{aligned}
R_T(a^*) &= \sum_{t=1}^{T} \mathop{\mathbf{E}}_{a \sim p_t} [\langle \ell_t, a - a^* \rangle] \\
&= \sum_{t=1}^{T} (\langle \ell_t, w_t - a^* \rangle + \langle \ell_t, w_t - \mu(p_t) \rangle).
\end{aligned} \tag{63}
$$

From the definition of bias given in (14), the part of $\langle \ell_t, w_t - a^* \rangle$ can be bounded as follows:

$$
\begin{aligned}
\langle \ell_t, w_t - a^* \rangle &= \mathop{\mathbf{E}}_{a_t, f_t} [\langle \ell_t - h_t(a_t, f_t), w_t - a^* \rangle + \langle h_t(a_t, f_t), w_t - a^* \rangle] \\
&\leq \mathrm{bias}(h_t, p_t, w_t, \ell_t) + \mathop{\mathbf{E}}_{a_t, f_t} [h_t(a_t, f_t), w_t - a^*].
\end{aligned} \tag{64}
$$

From this and (63), we have

$$
R_T(a^*) \leq \mathbf{E}\left[ \sum_{t=1}^{T} (\langle \ell_t, w_t - \mu(p_t) \rangle + \mathrm{bias}(h_t, p_t, w_t, \ell_t)) + \sum_{t=1}^{T} \langle h_t(a_t, f_t), w_t - a^* \rangle \right]. \tag{65}
$$

By a standard analysis of the FTRL framework (see, e.g., [Lattimore and Szepesvári, 2020a, Exercise 28.12]), we have

$$
\sum_{t=1}^{T} \langle h_t(a_t, f_t), w_t - a^* \rangle \leq \sum_{t=1}^{T} \frac{1}{\eta_t} \mathrm{stab}_\psi(\eta_t h_t(a_t, f_t), w_t) + \frac{1}{\eta_{T+1}} \psi(a^*). \tag{66}
$$

By combining (65) and (66), we obtain the bound of (14). $\square$

