# OpenReview forum: "An Exploration-by-Optimization Approach to Best of Both Worlds in Linear Bandits"
_NeurIPS.cc/2023/Conference — NeurIPS 2023 poster_

### Official Review · Reviewer_X2KX · 2023-06-30

**Soundness:** 3 good
**Presentation:** 3 good
**Contribution:** 3 good
**Rating:** 6
**Confidence:** 3

**Summary:**

The paper addresses the challenge of selecting an algorithm suited to the environment type, which is often unknown in real-world applications. This paper introduces the concept of best-of-both-worlds (BOBW) linear bandit algorithms that perform well in both stochastic and adversarial environments. Previous BOBW algorithms for linear bandit problems have achieved suboptimal regret bounds in stochastic environments due to an additional logarithmic factor. The authors use an existing approach called exploration by optimization (EXO), and prove that it achieves nearly optimal regret bounds in both regimes. EXO utilizes the exponential weight method to update the reference distribution and computes the sampling distribution and loss estimator to minimize an upper bound on regret. More precisely, the algorithm constructed using this approach achieves $O(d\sqrt{T \log T})$-regret in adversarial environments and $O(d^2 \log T/\Delta_{\min})$-regret in stochastic environments. The paper establishes a connection between the EXO approach and the SCRiBLe algorithm, which uses a follow-the-regularized-leader (FTRL) method with self-concordant barrier regularization. The authors also propose a variant called mean-oriented EXO that encompasses the EXO approach and allows for interpretation of existing SCRiBLe-type algorithms as EXO-based methods.

**Strengths:**

The paper addresses the challenge of constructing best-of-both-worlds linear bandit algorithms that perform well in both stochastic and adversarial environments. The distinction between these environments and the need for an optimal algorithm for both is considered as a quite significant open problem. The paper demonstrates the effectiveness of EXO in achieving nearly optimal regret bounds in both stochastic and adversarial environments by establishing regret bounds.

The paper also establishes a connection between the EXO approach and the SCRiBLe algorithm and its extensions. This connection helps in interpreting existing algorithms within the framework of exploration by optimization.

The paper suggests that the framework of exploration by optimization and best-of-both-worlds regret guarantees can be extended to other sequential decision-making problems beyond linear bandits. It mentions partial monitoring problems and episodic MDPs as potential areas for future research and highlights the broader implications of the EXO framework in various decision-making contexts.


**Weaknesses:**

While the paper presents an approach to constructing best-of-both-worlds linear bandit algorithms that perform well in both stochastic and adversarial environments, there are a few weak points that can be identified:

The paper assumes that the loss vectors are generated according to a specific model and that the environment falls under either a stochastic or adversarial setting. In real-world applications, the actual environment may not be strictly one or the other, and it might be challenging to accurately model the underlying process.

The paper primarily focuses on theoretical analysis and regret bounds, but it lacks an empirical evaluation or experimental results to validate the proposed algorithms. Without empirical evidence, it is challenging to assess the practical effectiveness and efficiency of the algorithms in real-world scenarios.

Improving a problem or solution by a log(T) term may not always be considered a significant breakthrough, depending on the context and the magnitude of the improvement.

The paper mentions that the proposed algorithms involve optimization problems (minimization problem (9)) and self-bounding techniques. While these techniques can lead to improved performance, they may introduce complexity and implementation challenges. The paper does not provide detailed insights into the practical feasibility and implementation aspects of the algorithms.

The paper does not explicitly discuss the limitations and assumptions of the proposed algorithms. It would be beneficial to address the assumptions made about the underlying environment, potential limitations of the algorithms, and scenarios where the algorithms may not perform optimally.


**Questions:**

Are there any limitations or potential challenges in implementing the EXO-based algorithm in real-world applications?

The paper mentions the use of the self-bounding technique for analyzing regret bounds in stochastic environments. Can you explain how this technique is applied and what advantages it offers?

How do you think the algorithm interpolate between the two regimes? For example, How does it perform in corrupted stochastic environments or environments with adversarial corruptions?

Can you elaborate on the potential extensions of the EXO framework to other sequential decision-making problems beyond linear bandits? How can this framework be applied to partial monitoring problems or episodic MDPs?

Are there any plans for future empirical studies?


**Limitations:**

The paper focuses solely on theoretical analysis.

The paper mentions that EXO can be complex in terms of structure and implementation. The details of how these algorithms can be efficiently implemented in practice are not provided, leaving open questions about their feasibility and scalability.

---

> ### Author Rebuttal · Authors · 2023-08-09
>
> We deeply appreciate the reviewers' thoughtful and comprehensive comments and feedback.
> The reviewers' insights have greatly contributed to the improvement of our work, and we sincerely appreciate the time and effort you have invested in this review.
> We hope our response below addresses your concerns.
>
> > Are there any limitations or potential challenges in implementing the EXO-based algorithm in real-world applications?
>
> Brief notes on the computational complexity and on the implementation of the algorithm can be found in Appendix E of the supplementary materials and in Remark 2.
> Unfortunately, it is not known if there is an efficient algorithm to solve the optimization problems (9) as we discussed in Appendix E,
> which is a major limitation in the implementation.
> However,
> we do not necessarily need to solve the optimization problem (9) exactly as noted in Remark 2.
> For example,
> it is expected that a polynomial-time algorithm that achieves regret bounds of Corollary 1 can be implemented given a separation oracle (or equivalently, a linear optimization oracle) for $\mathcal{A}$.
> Indeed,
> for achieving the regret bounds of Corollary 1,
> it suffices to find $p$ and $g$ such that $\Lambda_{q_t, \eta_t} (p, g)$ is bounded by the RHS of (10).
> The construction of such $p$ and $g$ is provided in the proof of Lemmas 2 and 3 (Lines 242--261),
> which can be performed using a separation oracle for $\mathcal{A}$.
> In fact,
> we can obtain samples from $p$ by using the techniques for log-concave sampling (e.g., [Lovász and Vempala, 2007]) and for computing convex combination expression (cf. Carathéodory's theorem for convex hull and [Schrijver, 1998, Corollary 14.1g]).
> However, the analysis of log-concave sampling and calculations of $H(p)$ (which is required for constructing $g$)
> including consideration of calculation errors can be highly complicated,
> and the computational cost can be very large, although on the order of polynomials (cf. [Hazan and Karnin, 2016, Corollary 6.2]).
> Therefore, it is also an important challenge to consider more efficient implementation methods.
> A more detailed discussion will be added in the revised version.
>
>
> > How do you think the algorithm interpolate between the two regimes? For example, How does it perform in corrupted stochastic environments or environments with adversarial corruptions?
>
> The proposed algorithm interpolates between the two regimes well.
> In fact,
> regret bounds in this paper can be applied to a class of stochastic environment with adversarial corruptions as noted on Lines 153--165.
> Discussion on Lines 153--165 and Theorem 1 together imply that
> the proposed approach achieves $O( \log T + \sqrt{C \log T} )$-regret
> (ignoring factors w.r.t. parameters other than $C$ and $T$)
> in stochastic environments with corruptions,
> where $C$ represents the total amount of corruption.
>
> > The paper mentions the use of the self-bounding technique for analyzing regret bounds in stochastic environments. Can you explain how this technique is applied and what advantages it offers?
>
> In the analysis based on the self-bounding technique,
> we exploit self-bounding constraints (2) to obtain regret bounds such that the regret itself appears on the right-hand side,
> e.g.,
> \\[
> 	R_T(a^*) \leq \sqrt{ \beta (R_T(a^*) + C) \log T } .
> \\]
> A similar expression can be found in Lines 524--525 in the supplementary material.
> This expression can be interpreted as a quadratic inequality for $R_T(a^*)$,
> and solving it yields $R_T(a^*) = O( \beta \log T + \sqrt{C \beta \log T}   )$.
> One advantage of this technique is that it also provides good bounds for stochastic environments with adversarial corruptions,
> i.e.,
> for the environments with $C > 0$.
>
> > Can you elaborate on the potential extensions of the EXO framework to other sequential decision-making problems beyond linear bandits? How can this framework be applied to partial monitoring problems or episodic MDPs?
>
> We believe that the algorithmic framework and Theorem 1 can be extended to other problems,
> such as partial monitoring problems or episodic MDPs,
> relatively easily.
> However, the analysis of $\Lambda^*$ provided in Section 4.3,
> which is essential for exploiting Theorem 1,
> heavily relies on a structure specific to linear bandits,
> and we believe that extending this part to other problems would require a great deal of nontrivial thought.
>
> > Are there any plans for future empirical studies?
>
> While we agree with the importance of empirical studies,
> we have no plans to conduct experimental studies in the near future,
> at this time.
> As we answered your first question,
> we need to consider how to improve the implementation of the proposed algorithm in order to make it work in a realistic computation time for high-dimensional problems.
> We believe that full-scale numerical experiments will become possible only after an efficient implementation method is found.
>
>
> References:
> - E. Hazan and Z. Karnin. Volumetric spanners: an efficient exploration basis for learning. The Journal of Machine Learning Research, 17(1):4062–4095, 2016.
> - L. Lovász and S. Vempala. The geometry of logconcave functions and sampling algorithms. Random Structures & Algorithms, 30(3):307–358, 2007.
> - A. Schrijver. Theory of Linear and Integer Programming. John Wiley & Sons, 1998.

---

> > ### Comment · Reviewer_X2KX · 2023-08-13
> >
> > I have read the other reviews and the rebuttal. I am satisfied with the answer provided by the authors. I will not change my score.

---

### Official Review · Reviewer_KqE4 · 2023-07-07

**Soundness:** 3 good
**Presentation:** 2 fair
**Contribution:** 2 fair
**Rating:** 5
**Confidence:** 3

**Summary:**

This paper considers the best-of-both-worlds problem for linear bandits. They investigate an Exploration-by-Optimization approach for this problem and show O(d^2\logT) regret for stochastic setting and O(d\sqrt{T}) regret for adversarial setting.

**Strengths:**

This paper investigates a new approach for Best-of-both-worlds problem in linear bandits.

In some special cases such as multi-armed bandits and hypercube, the authors provide improved results utilizing the problem structure.

Previous BoBW results mainly depend on FTRL-type algorithms, this work uses a new type of approach, which may be hoped to solve the BoBW problems with other structure.


**Weaknesses:**

Given existing works on BoBW problem for linear bandits, the convergence results provided by this paper are not optimal.

It is not clear what the advantage of EXO algorithm is compared with standard FTRL-type algorithms. Can you give more discussions?




**Questions:**

Please see the weakness part.

---

> ### Author Rebuttal · Authors · 2023-08-09
>
> Thank you for taking the time to read and review our paper.
> We hope the following answers address your questions.
>
> > Given existing works on BoBW problem for linear bandits, the convergence results provided by this paper are not optimal.
>
> As the reviewer pointed out, our bounds are not tight and we have positioned this as an important issue.
> Section 6 discusses possible approaches to these issues,
> and we intend to investigate the effectiveness of these approaches in future research.
>
> > It is not clear what the advantage of EXO algorithm is compared with standard FTRL-type algorithms. Can you give more discussions?
>
> Conventional FTRL-type algorithms for linear bandits,
> such as exponential weight algorithms with $p_t \approx q_t$ [Bubeck et al. 2012] and SCRiBLe with Dikin ellipsoid sampling [Abernethy et al., 2008, 2012],
> have been originally designed for adversarial environments,
> and do not achieve best-of-both-worlds regret bounds,
> i.e.,
> they do not have better bounds than $O(\sqrt{T})$ even in stochastic environments.
> The advantage of EXO algorithms compared to this is that they achieve best-of-both-worlds regret bounds,
> i.e.,
> they have $O(\log T)$-regret bounds for stochastic environments.
>
> References:
> - J. Abernethy, E. Hazan, and A. Rakhlin. Competing in the dark: An efficient algorithm for bandit linear optimization. In 21st Annual Conference on Learning Theory, 2008.
> - J. Abernethy, E. Hazan, and A. Rakhlin. Interior-point methods for full-information and bandit online learning. IEEE Transactions on Information Theory, 58(7):4164–4175, 2012.
> - S. Bubeck, N. Cesa-Bianchi, and S. M. Kakade. Towards minimax policies for online linear optimization with bandit feedback. In Conference on Learning Theory, pages 41–1. JMLR Workshop and Conference Proceedings, 2012.

---

> > ### Comment · Area_Chair_3hy9 · 2023-08-17
> > **Please engage in the rebuttal**
> >
> > Dear reviewer,
> > Please acknowledge the author's response and ideally tell us if that has changed your mind.

---

### Official Review · Reviewer_mre2 · 2023-07-07

**Soundness:** 3 good
**Presentation:** 3 good
**Contribution:** 3 good
**Rating:** 6
**Confidence:** 3

**Summary:**

The paper establishes $\log T$-style instance-dependent regret upper-bounds for linear bandit algorithms built upon the FTRL framework and appropriate exploration-by-optimization style update steps.

**Strengths:**

The $\log T$-style instance-dependent regret upper-bounds can be achieved on general convex action sets and a number of typical important specific action sets. The paper establishes the claimed results by first identifying a sufficient condition for the so-called self-bounding property for FTRL with exploration-by-optimization style update steps and thus $O(\log T)$ regret bounds (Theorem 1), then showing this condition is indeed satisfied under the continuous exponential weight methods.  Overall, the paper is clearly written.

**Weaknesses:**

Similar to most existing works on linear bandits that mainly discusses sample complexity instead the actual computational cost, the paper by default assumes one can effectively and accurately perform the computation of EXO update steps. However, for most convex action sets excepts for some special ones, these steps may be computationally intractable on an ordinary RAM machine. I would happy to see some discussion about the actual computation model (e.g., with oracle access to separation hyperplanes) and how to implement the proposed EXO algorithms.

**Questions:**

See the weaknesses part.


**Limitations:**

See the weaknesses part.

---

> ### Author Rebuttal · Authors · 2023-08-09
>
> Thank you for taking the time to review our paper.
> We hope that our response below addresses your concerns.
>
> > Similar to most existing works on linear bandits that mainly discusses sample complexity instead the actual computational cost, the paper by default assumes one can effectively and accurately perform the computation of EXO update steps. However, for most convex action sets excepts for some special ones, these steps may be computationally intractable on an ordinary RAM machine. I would happy to see some discussion about the actual computation model (e.g., with oracle access to separation hyperplanes) and how to implement the proposed EXO algorithms.
>
> Brief notes on the computational complexity and on the implementation of the algorithm can be found in Appendix E of the supplementary material and in Remark 2.
> We consider that one reasonable computation model would be the Real-RAM model equipped with a separation oracle for (the convex hull of) the action set $\mathcal{A}$ as the reviewer suggested.
> Unfortunately, it is not known if there is an efficient algorithm to solve the optimization problems (9) as we discussed in Appendix E.
> However,
> we do not necessarily need to solve the optimization problem (9) exactly as noted in Remark 2.
> For example,
> it is expected that a polynomial-time algorithm that achieves regret bounds of Corollary 1 can be implemented given a separation oracle (or equivalently, a linear optimization oracle) for $\mathcal{A}$.
> Indeed,
> for achieving the regret bounds of Corollary 1,
> it suffices to find $p$ and $g$ such that $\Lambda_{q_t, \eta_t} (p, g)$ is bounded by the RHS of (10).
> The construction of such $p$ and $g$ is provided in the proof of Lemmas 2 and 3 (Lines 242--261),
> which can be performed using a separation oracle for $\mathcal{A}$.
> In fact,
> we can obtain samples from $p$ by using the techniques for log-concave sampling (e.g., [Lovász and Vempala, 2007]) and for computing convex combination expression (cf. Carathéodory's theorem for convex hull and [Schrijver, 1998, Corollary 14.1g]).
> However, the analysis of log-concave sampling and calculations of $H(p)$ (which is required for constructing $g$)
> including consideration of calculation errors can be highly complicated,
> and the computational cost can be very large, although on the order of polynomials (cf. [Hazan and Karnin, 2016, Corollary 6.2]).
> A more detailed discussion will be added in the revised version.
>
> References:
> - E. Hazan and Z. Karnin. Volumetric spanners: an efficient exploration basis for learning. The Journal of Machine Learning Research, 17(1):4062–4095, 2016.
> - L. Lovász and S. Vempala. The geometry of logconcave functions and sampling algorithms. Random Structures & Algorithms, 30(3):307–358, 2007.
> - A. Schrijver. Theory of Linear and Integer Programming. John Wiley & Sons, 1998.

---

> > ### Comment · Reviewer_mre2 · 2023-08-15
> > **Thanks for the response**
> >
> > Thanks for the response. The discussion looks good to me. I will keep my score.

---

### Official Review · Reviewer_JH8B · 2023-07-10

**Soundness:** 3 good
**Presentation:** 3 good
**Contribution:** 3 good
**Rating:** 7
**Confidence:** 4

**Summary:**

This paper studies the "best-of-both-world" problem for linear bandits, meaning that designing a single algorithm that simultaneously achieves a $O(\sqrt{T\log T})$ regret in the adversarial case and $O(\log T)$ regret in the stochastic case. They propose to use the Exploration-by-Optimization approach to obtain such bounds. The algorithm can be viewed as a continuous exponential weight algorithm with an optimally designed action sampling scheme.

**Strengths:**

- The approach provides a unified and insightful way to obtain near-optimal best-of-both-worlds bounds for linear bandits and several of its special cases. The algorithm is conceptually simpler than previous work such as [Lee et al., 2021] and [Dann et al., 2023].
- The algorithm is based on a general framework of Exploration-by-Optimization, which has been proven to work for general adversarial decision making problems. Therefore, the proposed method has potential to be extended to more general cases, though the current analysis is only for linear bandits.
- There are several novel elements in the analysis, especially Lemma 3. It shows that continuous exponential weights with a simple sampling scheme already works well.

**Weaknesses:**

- The analysis relies on the unique optimal action assumption, and the bound in the stochastic environment can be arbitrarily worse than the  instance-dependent bound by Lattimore and Szepesvari (2017). However, these are general open questions in the field but not the specific issues of this work.

**Questions:**

See the above sections.

**Limitations:**

There is no potential negative societal impact.

---

> ### Author Rebuttal · Authors · 2023-08-09
>
> We sincerely appreciate the time and effort you have invested in this review.
>
> > The analysis relies on the unique optimal action assumption, and the bound in the stochastic environment can be arbitrarily worse than the instance-dependent bound by Lattimore and Szepesvari (2017). However, these are general open questions in the field but not the specific issues of this work.
>
> As discussed in Section 6, we recognize the two points you raised as important issues.
> Section 6 also discusses possible approaches to these issues,
> and we intend to investigate the effectiveness of these approaches in future research.

---

> > ### Comment · Reviewer_JH8B · 2023-08-17
> >
> > Thanks for the answer. I think this is a good work, and I keep my positive scoring.

---

### Decision · Program_Chairs · 2023-09-21

**Decision:**

Accept (poster)

**Comment:**

This paper provides a novel solution to the Best of both worlds problem for linear bandits. It is the first paper using a recent exploration by optimization approach for BoB problems and the techniques are non-trivial and potentially useful for future work.
The algorithm does not obtain better bounds than prior work, but is simpler and arguably more elegant. The work has the same weaknesses, i.e. uniqueness assumption of the best arm and sub-optimal asymptotic instance dependent bound, as the literature.
Unlike the state-of-the-art solution for this problem, the algorithm does not rely on restarts and doubling and the uniqueness assumption might be an artefact of the analysis.

There is a consensus among the reviewer that this paper is above the acceptance threshold, but the lack of improvement over the sota and the lack of an implementable algorithm do not allow for a higher than poster rating.